# DISCO: DIVERSIFYING SAMPLE CONDENSATION FOR EFFICIENT MODEL EVALUATION

Alexander Rubinstein[1]     Benjamin Raible[1]     Martin Gubri[2]     Seong Joon Oh[1]

[1]Tübingen AI Center, University of Tübingen
[2]Parameter Lab

🌐 Project Page     ⌸ DISCO Codebase

## ABSTRACT

Evaluating modern machine learning models has become prohibitively expensive. Benchmarks such as LMMs-Eval and HELM demand thousands of GPU hours per model. Costly evaluation reduces inclusivity, slows the cycle of innovation, and worsens environmental impact. To address the growing cost of standard evaluation, new methods focused on efficient evaluation have started to appear. The typical approach follows two steps. First, select an anchor subset of data. Second, train a mapping from the accuracy on this subset to the final test result. The drawback is that anchor selection depends on clustering, which can be complex and sensitive to design choices. We argue that promoting diversity among samples is not essential; what matters is to select samples that *maximise diversity in model responses*. Our method, **Diversifying Sample Condensation** (**DISCO**), selects the top-k samples with the greatest model disagreements. This uses greedy, sample-wise statistics rather than global clustering. The approach is conceptually simpler. From a theoretical view, inter-model disagreement provides an information-theoretically optimal rule for such greedy selection. **DISCO** shows empirical gains over prior methods, achieving state-of-the-art results in performance prediction across MMLU, Hellaswag, Winogrande, and ARC.

## 1 INTRODUCTION

Model evaluation is becoming increasingly costly. Models have grown in size, which makes each inference expensive. Recent scaling of test-time computation has further raised the cost per task. End-user requirements have also broadened, covering both the content of the output and its style (Wang et al., 2018; Liang et al., 2022; Kim et al., 2023; Zhang et al., 2024). As a result, evaluation on modern benchmarks often requires hundreds to thousands of GPU hours. For instance, LMMs-Eval can take between 30 and 1400 hours on 8×A100 GPUs (Zhang et al., 2024). HELM requires more than 4000 GPU hours (Liang et al., 2022).

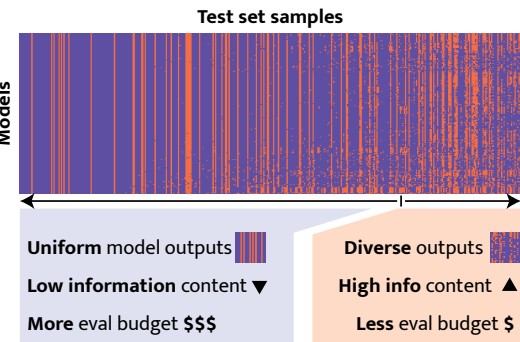

**Test set samples**

Figure 1: **Imbalance**. More evaluation budget is spent on less informative samples in test sets.

Several efficient evaluation approaches have emerged. A common framework works in two parts: subset selection and performance prediction. The first part selects a static subset of anchor points from the evaluation dataset. The second part predicts full benchmark performance by extrapolating from accuracy on this subset. To select anchor points, existing methods often rely on clustering. Samples are grouped by the similarity of responses they induce in a set of reference models (Vivek et al., 2023; Polo et al., 2024). Variants of this framework include dynamic anchor selection (Hofmann et al., 2025), modified prediction models (Kipnis et al., 2024), and new benchmarks for method comparison (Zhang et al., 2025).

We seek to improve both parts of this framework. For subset selection, we argue that diversity among samples is not essential. What matters is **diversity in model responses**. We prove that inter-model disagreement is the most informative signal for estimating benchmark performance when the goal is to differentiate and rank models (Proposition 1). Evaluation should therefore focus on samples that elicit varied responses (Figure 1). For performance prediction, we argue that existing methods add unnecessary complexity by estimating hidden model parameters before predicting test performance (Polo et al., 2024; Kipnis et al., 2024). We instead propose a direct route. Model signatures, defined as the concatenation of outputs on the selected subset, serve as inputs to simple predictors of benchmark performance. This framework is simpler, yet matches and surpasses more complex alternatives.

We validate these ideas through **Diversifying Sample Condensation** (**DISCO**). DISCO selects a small, informative subset of evaluation samples by focusing on model disagreement. Disagreement is measured by predictive diversity scoring (PDS, Rubinstein et al. (2024)), originally proposed for out-of-distribution detection. A simple metamodel then predicts benchmark performance directly from the model signatures on this subset. We evaluate DISCO in both language and vision domains. On MMLU, for example, DISCO reduces evaluation cost by 99.3% (see Appendix B.4) with only 1.07 percentage points of error. Compared with prior methods such as Anchor Points (Vivek et al., 2023), TinyBenchmarks (Polo et al., 2024), and Metabench (Kipnis et al., 2024), DISCO achieves a stronger efficiency–precision trade-off.

## 2 RELATED WORK

We review prior work relevant to our approach. We first highlight the escalating cost of evaluation for contemporary large models and motivate the need for efficiency. We then survey prior attempts at efficient benchmarking, covering instance and task reduction techniques. Finally, we describe our novelty and contributions.

**Cost of evaluation.** The evaluation of modern large models is currently driven by increasingly sophisticated benchmarks assessing a wide array of capabilities, from the foundational GLUE (Wang et al., 2018) and the comprehensive HELM (Liang et al., 2022) to LMMs-Eval for multimodal models (Zhang et al., 2024), the diverse BIG-bench (Srivastava et al., 2022), Prometheus for measuring diverse LLM capabilities (Kim et al., 2023), and GAIA for general AI assistants (Mialon et al., 2023). This progress comes at an escalating cost: models have grown significantly in size, making each inference step more resource-intensive, while the scaling of test-time computations has dramatically increased the per-task evaluation costs. Furthermore, end-user requirements have diversified to encompass not only output content but also style and manner. Consequently, a single evaluation on modern benchmarks can demand hundreds to thousands of GPU hours. For example, LMMs-Eval can require between 30 and 1400 hours on 8×A100 GPUs per model (Zhang et al., 2024; Polo et al., 2024), and HELM evaluations can exceed 4000 GPU hours per model (Liang et al., 2022; Polo et al., 2024).

**Label-efficient evaluation.** In the pre-LLM context, labelling a test set used to be a cost bottleneck for evaluation. In this context, the concept of "active testing" has been explored, where labelling budget is maximally assigned to information-rich samples (Majumdar & Niksic, 2017; Ji et al., 2021; Deng & Zheng, 2021; Kossen et al., 2021; Hu et al., 2023; Kossen et al., 2022; Huang et al., 2024; Fogliato et al., 2024). In our case, we are concerned with the *inference costs* of evaluation. As such, active testing approaches are not directly applicable, as they require a full inference over the test set to identify informative samples to label.

**Efficient benchmarking.** In the LLM era, benchmarks have diversified to measure multiple capabilities and styles of model behaviours. Researchers have proposed strategies to build an efficient benchmark in the first place (Perlitz et al., 2023; Rädsch et al., 2025). There were attempts to compress multiple benchmarks, measuring an array of capabilities of LLMs, into a single one by eliminating redundancies (Kipnis et al., 2024; Zhao et al., 2024; Yuan et al., 2025). Others have focused on selection of small, informative subsets, also known as "Anchor point" approaches (Vivek et al., 2023; Polo et al., 2024; Li et al., 2025; Gupta et al., 2025). Given an entire dataset, they compute a small subset of data points according to the *representativeness* criterion, determined through the correctness patterns of a large number of *source models*. Subsequently, target model performance is estimated based on weighted accuracy computed on the selected subset. In particular, tinyBenchmarks (Polo et al., 2024) have adopted Item Response Theory (IRT) (Lord & Novick, 2008)

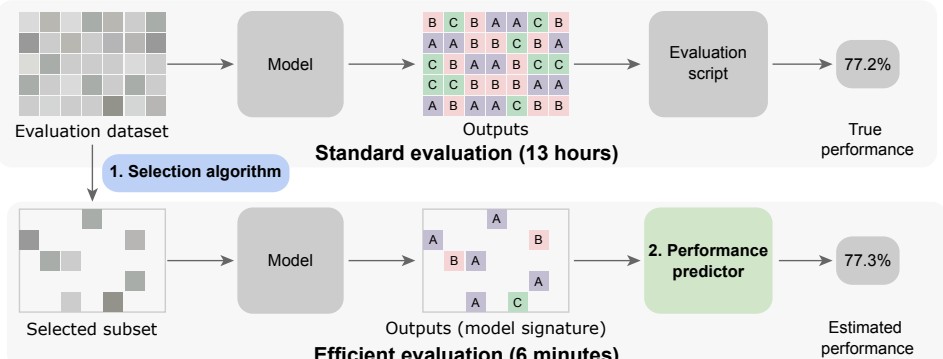

Figure 2: **Problem overview**. We aim at selecting a much smaller evaluation dataset than the original evaluation dataset, while keeping the estimated performances as close as possible. Figure 3 details the selection algorithm and the performance predictor.

to estimate model performance in a principled manner. Hofmann et al. (2025) proposed an IRT-based approach to LLM evaluation that selects anchor points dynamically for each model, guided by its predictions on previously chosen anchors. To address the growing number of methods for efficient LLM evaluation, Zhang et al. (2025) recently introduced a large-scale benchmark. In this work, we adopt approaches from the black-box model analysis techniques, explained below.

**Our novelty and contribution.** We differentiate our approach, Diversifying Sample Condensation (DISCO), from previous work in two aspects. (1) *model disagreement* (Rubinstein et al., 2024) is a simpler and more effective proxy for sample informativeness than *representativeness* (Vivek et al., 2023; Polo et al., 2024). (2) The application of metamodels on model signatures is a simpler and more effective approach than direct accuracy evaluation approaches (Vivek et al., 2023) or prior approaches that require estimating latent model parameters (Polo et al., 2024; Kipnis et al., 2024).

## 3 PROBLEM

Our task is the estimation of model performance on a benchmark. Let $f : \mathcal{X} \rightarrow \mathcal{Y}$ be a predictive model over a dataset $\mathcal{D} := \{(x_1, y_1), \ldots, (x_N, y_N)\}$ sampled iid from some distribution. We are interested in estimating the model performance on the dataset $S_{\mathcal{D}}^f$. An example metric for model performance is accuracy: for a probabilistic classifier $f : \mathcal{X} \rightarrow [0, 1]^C$, accuracy is defined as $\frac{1}{N} \sum_i \mathbf{1}\{\arg\max_c f_c(x_i) = y_i\}$.

We are interested in estimating $S_{\mathcal{D}}^f$ in a cost-effective way. We seek ways to sample a subset of size $K \ll N$ from the original set $\mathcal{D}$ to estimate $S_{\mathcal{D}}^f$. The overall problem is described in Figure 2. An integral ingredient for both prior work and ours is the set of *source models* $\mathcal{F} = \{f^1, \ldots, f^M\}$, a held-out set of models whose ground-truth performances are known. We define the *target models* $\tilde{\mathcal{F}} = \{\tilde{f}^1, \ldots, \tilde{f}^{M_{\text{target}}}\}$ as the models whose performances we aim to estimate.

## 4 SOLUTION

This section presents DISCO, our solution to the problem of efficient performance evaluation. DISCO is composed of two steps: *(i)* the dataset selection, where given an original dataset and an held-out set of source models, we identify a much smaller subset of samples; and *(ii)* the performance prediction, where given the model outputs on our DISCO selected evaluation set, we estimate the model performance on the original set.

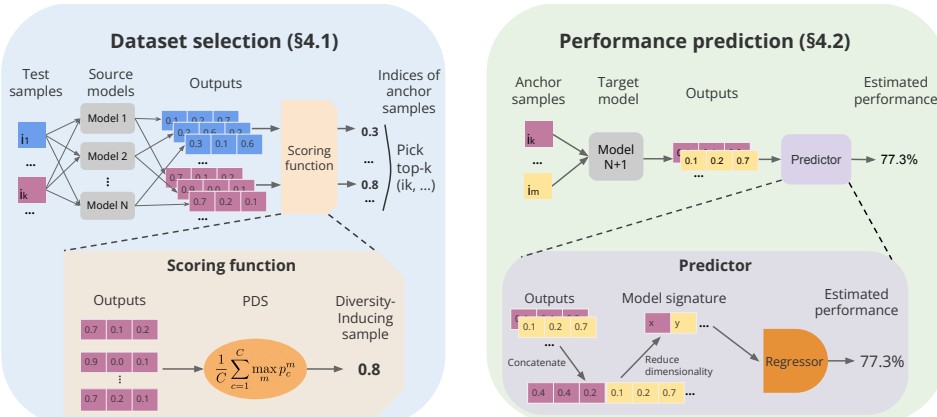

Figure 3: **DISCO overview**. First, we select a subset of an evaluation dataset with the most informative samples. Second, we predict the performance of unseen models from their outputs on the selected samples.

## 4.1 DATASET SELECTION

At this stage, we require a score that quantifies each sample's informativeness for predicting performance on the full dataset. Using this score, we rank the samples and select a top-k subset that best preserves the dataset's information content.

### 4.1.1 PRIOR SELECTION METHODS

We first review existing approaches for selecting representative data points in the evaluation set, referred to as anchor points.

*Anchor-conf* Vivek et al. (2023) choose $K$ anchors $A = \{a_k\}_1^K \subset \{x_i\}_1^N$ that minimise the sum of distances between each data point and the closest anchor: $\min_A \sum_{i,k} d\left(e(x_i), e(a_k)\right)$, where the $e(x)$ abbreviates the concatenated model likelihoods $e(x, y) := \left[f^1(x)_y, \dots, f^M(x)_y\right]$ for the input and ground-truth label $(x, y)$ for the source models $\mathcal{F} = \{f^1, \dots, f^M\}$.

*Anchor-corr* (Polo et al., 2024) is nearly identical to *Anchor-conf*, except that the embedding uses correctness scores instead of likelihoods: $e(x, y) := \{s^1(x, y), \dots, s^M(x, y)\}$, where $s^m(x, y) := 1(\arg\max_c f^m(x)_c = y)$ encodes correctness of model $f^m$ on sample $x$.

*Anchor-IRT* (Polo et al., 2024) uses the Item-Response Theory (IRT) to define a parametric model $\Pr\left(s_i^m = 1 \mid \theta^m, \alpha_i, \beta_i\right) = \text{sigmoid}(-\alpha_i^\top \theta^m + \beta_i)$. It predicts the correctness of a model $f^m$ on a sample $x_i$ with parameters $\theta^m \in \mathbb{R}^d$, $\alpha_i \in \mathbb{R}^d$, and $\beta_i \in \mathbb{R}$. Using observations of the sample-wise correctness of source models $(x_i, y_i, s_i^m)$, the parameters are inferred with an Expectation-Maximisation algorithm. Now, they continue the anchor selection based on the sample-wise embeddings $e(x_i) := (\alpha_i, \beta_i)$.

*Best for validation* Kipnis et al. (2024) finds an anchor set $A$ through an iterative search. The algorithm first generates a large number of candidate anchor sets, $\{A_1, \dots, A_P\}$, by uniformly sampling from the full dataset $\mathcal{D}$. For each candidate set $A_p$, a simple scalar-to-scalar regression model, $g_p$, is trained on the source models $\mathcal{F}$. This model learns to map the performance on the subset, $S_{A_p}^f$, to the known ground-truth performance on the full dataset, $S_{\mathcal{D}}^f$. Each trained regressor $g_p$ is subsequently evaluated on a held-out validation set of models. The final anchor set $A$ is selected as the candidate $A_p$ whose corresponding regressor $g_p$ yields the lowest prediction error (e.g., RMSE) on this validation set.

**How DISCO differs.** Unlike clustering, we use sample-wise statistics to determine samples with maximal information content. This greatly simplifies the sampling procedure. We exploit the **model diversity**, not **model confidence** or **correctness**. A set of models can be highly confident and diverse at the same time. We argue that inputs that induce model diversity are more useful for performance prediction.

### 4.1.2 DISCO SELECTION

We now present our selection method. In this part, we explain how we identify such samples in the test dataset. Our sample selection strategies are illustrated in Figure 3. The main approach in **Diversifying Sample Condensation (DISCO)** is to select a subset $\mathcal{D}_{\text{DISCO}}$ of the original evaluation set $\mathcal{D}$ by sampling the top-k samples based on disagreement score, such as PDS. This follows the intuition shown in Figure 1.

We start with an information-theoretic observation below.

**Proposition 1.** *Let $\mathcal{D} = \{(x_i, y_i)\}_i^N$ be a test set and $m \sim \text{Unif}\{1, \ldots, M\}$ (A1) be the index of a uniformly chosen model. Let $f_c^m(x_i) \in [0, 1]$ be the predictive probability for class $c$ of model $f^m$ on input $x_i$. We write $\widehat{y}_i^m$ for the categorical random variable following $\text{Cat}(f_1^m(x_i), \ldots, f_C^m(x_i))$. Define ensemble mean prediction to be $\bar{f}_c(x_i) := \mathbb{E}_m[f_c^m(x_i)]$ for each class $c$ and define corresponding prediction random variable as $\widehat{y}_i$ following $\text{Cat}(\bar{f}_1(x_i), \ldots, \bar{f}_C(x_i))$. Let $S(m) = S(f^m, \mathcal{D})$ denote a function of model $m$ and dataset $\mathcal{D}$, such as model accuracy, that is injective with respect to $m$. Assume that the only randomness in $\widehat{y}^m$ comes from $m$ (A2). Then,*

$$\text{MI}_{m,\widehat{y}_i}\left(S(m); \widehat{y}_i\right) = H(\widehat{y}_i) - \mathbb{E}_m\left[H\left(\widehat{y}_i^m\right)\right] = \text{JSD}\left(\widehat{y}_i^1, \ldots, \widehat{y}_i^M\right).$$

*where $H(\cdot)$ is entropy, $\text{MI}(\cdot)$ is mutual information, and $\text{JSD}(\cdot)$ is generalised Jensen-Shannon Divergence for multiple distributions (Fuglede & Topsoe, 2004).*

*See proof in Appendix G.*

We conclude that the sample $i$ conveying the greatest level of information for the prediction of $S(m)$ (e.g., model accuracy) is the one with the greatest $\text{JSD}\left(\widehat{y}_i^1, \ldots, \widehat{y}_i^M\right)$. This generalised Jensen-Shannon divergence translates to the diversity of distributions (Fuglede & Topsoe, 2004). Based on the insight that model diversity matters for performance prediction, we also consider an alternative measure that measures the model diversity: predictive diversity score (PDS) (Rubinstein et al., 2024). It is more interpretable, as it is a continuous generalisation of the number of unique argmax category predictions among $M$ source models:

$$\text{PDS}\left(\widehat{y}_i^1, \ldots, \widehat{y}_i^M\right) := \frac{1}{C} \sum_c \max_m f_c^m(x_i). \tag{1}$$

PDS is related to JSD through the enveloping inequalities below:

**Proposition 2.** *Denoting $\text{PDS}_i := \text{PDS}\left(\widehat{y}_i^1, \ldots, \widehat{y}_i^M\right), \text{JSD}_i := \text{JSD}\left(\widehat{y}_i^1, \ldots, \widehat{y}_i^M\right)$ for each sample $i$, we have*

$$\frac{2}{M^2 \ln 2}\left(\text{PDS}_i - 1\right)^2 \leqslant \text{JSD}_i \leqslant \frac{M}{M-1}\log M \cdot \left(\text{PDS}_i - 1\right).$$

*See proof in Appendix H.3.*

In the experiments, we consider both JSD and PDS as criteria for sample selection.

## 4.2 PERFORMANCE PREDICTION

Once a subset of dataset samples $A$ is selected, we use the responses of the target model $f$ on $A$ to estimate the true performance.

### 4.2.1 PRIOR PREDICTION METHODS

We first review existing approaches for estimating the true performance using predictions on anchor points $A = \{a_1, \ldots, a_K\}$.

*Weighted sum* Vivek et al. (2023) estimates the true performance by directly computing the accuracy on the anchor set: $\text{WS}(f, A) := (1/K) \sum_k w_k s_k^m$, where $w_k$ is the number of original training samples $x_i$ assigned to the anchor $a_k$ in the *Anchor-Corr* method.

*p-IRT* (Polo et al., 2024): makes adjustments to the vanilla accuracy on the anchor set by adding a correction term derived from the IRT in *Anchor-IRT* in: $\text{p-IRT}(f, A) := (1/K) \sum_{k \in A} s_k +$

$1/(N - K) \sum_{k \notin A} p_i$, where $\hat{p}_i$ is the IRT estimation computed based on the parameters obtained in *Anchor-IRT*.

*gp-IRT* (Polo et al., 2024) is a mixture of the two approaches above: $\text{gp-IRT}(f, A) = \lambda \cdot \text{WS}(f, A) + (1 - \lambda) \cdot \text{p-IRT}(f, A)$ where $\lambda \in [0, 1]$.

*ability-IRT* Kipnis et al. (2024) is a two-stage method that uses the anchor set $A$ as a diagnostic tool rather than just a miniature test. First, it uses a pre-calibrated IRT model to estimate a latent "ability" score, $\hat{\theta}^f$, from the target model's pattern of correct and incorrect responses on $A$. Second, a pre-trained regressor, $g$, predicts the final performance $S_{\mathcal{D}}^f$ using both the simple anchor set accuracy $\hat{S}_A^f$ and this more informative ability score $\hat{\theta}^f$ as input features. The final prediction is given by $S_{\mathcal{D}}^f = g(\hat{S}_A^f, \hat{\theta}^f)$, leveraging a deeper measure of the model's capability to improve the estimate.

**How DISCO differs.** Previous prediction methods rely on scalar summaries of performance, such as the (weighted or corrected) accuracy on the anchor set. In contrast, our approach leverages a much richer signal: the **model signature**, defined as the concatenation of the model's raw outputs on the selected samples. By learning a direct mapping from the high-dimensional signature to the final performance, we bypass the complexities of psychometric modeling and demonstrate that a simpler, more direct approach can be more effective.

### 4.2.2 DISCO PREDICTION

Given a smaller set of test dataset $\mathcal{D}_{\text{DISCO}}$, we estimate the performance of a model $f$ as closely as possible to the true full test performance $S_{\mathcal{D}}^f$. We deliberately opt for simple approaches here, in order to make a point that simple is best; we also compare against a rather complex prior work and show that our simple method outperforms it. Our performance prediction framework is depicted in Figure 3.

**Model signatures.** We hypothesise that models with similar output patterns on $\mathcal{D}_{\text{DISCO}}$ will exhibit similar performance. To capture this pattern, we define a **model signature** as the concatenation of the model's outputs on $\mathcal{D}_{\text{DISCO}}$: $f(\mathcal{D}_{\text{DISCO}}) := [f(x_1), \ldots, f(x_L)]$.

Such a function signature may have high dimensionality, as it is the product of model output dimensionality (e.g., 1000 for ImageNet) and the number of selected samples $|\mathcal{D}_{\text{DISCO}}|$ (e.g., can go up to 50k for ImageNet validation set). To reduce the storage burden and improve generalizability, we consider applying a dimensionality reduction technique based on principal component analysis (PCA): $Q \circ f(\mathcal{D}_{\text{DISCO}})$.

**KNN prediction.** Built on the hypothesis that the similarities in function signature imply performance similarity, we consider the kNN predictor based on a held-out set of models $\mathcal{F}$. Given a function $f$ to evaluate, we identify the K most similar models in $\mathcal{F}$ using the Euclidean distance between their signatures after dimensionality reduction. We estimate $f$'s performance by averaging the performances of the K most similar models.

**Parametric mapping.** We also consider a parametric prediction variant. A single parametric mapping $R$ is trained for the prediction of model performance. As the training set, we use $M$ model signatures $Q \circ f_1(\mathcal{D}_{\text{DISCO}}), \ldots Q \circ f_M(\mathcal{D}_{\text{DISCO}})$ for $\mathcal{F}$ as the training set for the regression problem of training a mapping $R(\cdot)$ to let $R \circ Q \circ f_m(\mathcal{D}_{\text{DISCO}})$ approximate $\hat{S}_{\mathcal{D}}^f$. The predictor $R$ can be implemented using a neural network, linear regression, or a Random Forest, for example.

## 5 EXPERIMENTS

In this section, we introduce the evaluation protocol (§5.1) and the experimental setup (§5.2), present the main results of Diversifying Sample Condensation (DISCO) in language domain (§5.3), analyse contributing factors (§5.4), and demonstrate that the method is domain-agnostic and can also be successfully applied to the vision domain (§5.5).

## 5.1 Evaluation protocol

To ensure a fair comparison, all methods follow an identical evaluation protocol, i.e., they use the same ingredients and perform the same sequence of steps during the training and testing stages.

**Training**. Following § 4, select anchor datapoints and train the performance predictor.

Input: source models $\mathcal{F} = \{f^1, \dots, f^M\}$, full test dataset $\mathcal{D} = (\mathcal{D}_x, \mathcal{D}_y)$ with questions $\mathcal{D}_x$ and ground truth answers $\mathcal{D}_y$, parameter $K$.

Output: set of anchor datapoints $A_K$, predictor $R$.

1. Evaluate source models $\mathcal{F}$ on $\mathcal{D}_x$ and obtain model outputs
   $\mathcal{F}(\mathcal{D}_x) = \{f(x) : x \in \mathcal{D}_x, f \in \mathcal{F}\}$.
2. Calculate their full test performance (e.g., accuracy) $S_{\mathcal{D}}^{\mathcal{F}} = \{S_{\mathcal{D}}^f : f \in \mathcal{F}\}$ based on $\mathcal{F}(\mathcal{D}_x)$ and $\mathcal{D}_y$.
3. Use $\mathcal{F}(\mathcal{D}_x)$ and optionally $\mathcal{D}_y$ to select a set $A_K \subseteq \mathcal{D}_x$ of $K$ anchor datapoints with respective selection method (e.g., PDS/JSD, IRT, etc.) explained in § 4.1.
4. Train a predictor $R$ (e.g., Random Forest, gp-IRT, ability-IRT, etc.) explained in § 4.2 to predict full test performance from model's outputs on anchor datapoints
   $f(A_K) = \{f(x) : x \in A_K\}$, such that $S_D^f \approx \widehat{S}_D^f = R(f(A_K)), \forall f \in \mathcal{F}$.

**Testing**. Test the performance predictor that is trained as explained above.

Input: target models $\tilde{\mathcal{F}} = \{\tilde{f}^1, \dots, \tilde{f}^{M_{\text{target}}}\}$, set of anchor points $A_K$, predictor $R$, ground truth performances of target models $S_{\mathcal{D}}^{\tilde{\mathcal{F}}} = \{S_{\mathcal{D}}^f : f \in \tilde{\mathcal{F}}\}$ computed the same way as $S_{\mathcal{D}}^{\mathcal{F}}$.

Output: performance of the efficient evaluation method.

1. Evaluate target models $\tilde{\mathcal{F}}$ on anchor points $A_K$ and obtain their outputs
   $\tilde{\mathcal{F}}(A_K) = \{f(x) : x \in A_K, f \in \tilde{\mathcal{F}}\}$.
2. Use predictor $R$ to estimate the ground truth performances of the target models
   $\widehat{S}_{\mathcal{D}}^{\tilde{\mathcal{F}}} = \{R(f(A_K)) : f \in \tilde{\mathcal{F}}\}$.
3. Calculate performance (e.g., MAE, Spearman rank correlation, etc.) of the efficient evaluation method by comparing $S_{\mathcal{D}}^{\tilde{\mathcal{F}}}$ and $\widehat{S}_{\mathcal{D}}^{\tilde{\mathcal{F}}}$.

## 5.2 Setup

We describe our experimental setup: the datasets, metrics, models, and model splits.

**Datasets.** We evaluate DISCO on four widely used language modeling benchmarks: MMLU (Hendrycks et al., 2021), HellaSwag (Zellers et al., 2019), Winogrande (Sakaguchi et al., 2021), and ARC (Clark et al., 2018). Details on the benchmarks can be seen in Appendix A.

**Metrics.** We evaluate DISCO and baseline approaches using two complementary metrics. First, the Mean Absolute Error (*MAE*) of the model accuracies, reported as percentage points (%p), captures the absolute error of accuracy prediction. Second, to assess the consistency of the relative ordering of models, we report the Spearman rank correlation (*Rank*) in model ranking between the true and estimated model performances.

**Models.** Building on the TinyBenchmarks framework (Polo et al., 2024), we evaluate 424 large language models (LLMs) from Hugging Face's Open LLM Leaderboard (Fourrier et al., 2024). The models cover GPT- (Radford et al., 2019), LLaMA- (Touvron et al., 2023), DeepSeek- (DeepSeek-AI et al., 2025), and BERT-style (Devlin et al., 2019) architectures, with model sizes ranging from 1.3 billion to 72 billion parameters.

**Model split.** DISCO is based on a meta-model approach where a predictor is constructed based on the model signatures of a pool of source models $\mathcal{F}$ and tested on a disjoint set of target models. This approach has traditionally been criticised for its dependency on the set of existing models: the predictor may fail to retain performance with unforeseen changes in future models. To address

| Approach | Selection | Prediction | MMLU (14k) | | HS (10k) | | WG (1.3k) | | ARC (1.2k) | |
|---|---|---|---|---|---|---|---|---|---|---|
| | §4.1 | §4.2 | MAE↓ | Rank↑ | MAE↓ | Rank↑ | MAE↓ | Rank↑ | MAE↓ | Rank↑ |
| Baseline | Random | Direct eval. | 3.45 | 0.916 | 2.85 | 0.839 | 3.60 | 0.827 | 2.61 | 0.898 |
| tinyBenchmarks | Random | gp-IRT | 2.79 | 0.922 | 1.96 | 0.819 | 1.64 | 0.928 | 2.22 | 0.921 |
| | Anchor-IRT | gp-IRT | 3.25 | 0.922 | 2.19 | 0.830 | 2.24 | 0.850 | 4.55 | 0.708 |
| | Anchor-corr | gp-IRT | 2.08 | 0.927 | 1.27 | 0.937 | 1.95 | 0.918 | 2.18 | 0.948 |
| Metabench | Best for val. | ability-IRT | 2.08† | 0.904† | **0.80†** | 0.974† | 1.23† | 0.947† | **1.14†** | **0.971†** |
| Model signature | Random | Sig. + kNN | 1.82 | 0.912 | 1.49 | 0.899 | 1.58 | 0.920 | 2.30 | 0.905 |
| | | Sig. + RF | 1.81 | 0.933 | 1.36 | 0.938 | 1.29 | 0.926 | 1.72 | 0.938 |
| DISCO (ours) | High PDS | Sig. + kNN | 1.31 | 0.972 | 1.32 | 0.956 | 1.19 | 0.951 | 1.96 | 0.937 |
| | | Sig. + RF | **1.07** | **0.987** | 1.01 | **0.984** | **1.00** | 0.967 | 1.47 | **0.971** |
| | High JSD | Sig. + kNN | 1.14 | 0.975 | 1.50 | 0.944 | 1.26 | 0.955 | 2.11 | 0.939 |
| | | Sig. + RF | 1.30 | **0.987** | 0.86 | 0.972 | 1.09 | **0.973** | 1.75 | 0.938 |

Table 1: **DISCO achieves state-of-the-art test-set compression by using model signatures combined with PDS for accurate performance prediction**. Compression of MMLU, HellaSwag (HS), Winogrande (WG), and ARC datasets by DISCO (ours), tinyBenchmarks, Metabench, and other baselines. For each dataset, we reduce the test set to 100 data points (except for Metabench, see below), achieving inference cost reduction of 99.3% and 99.0%, on MMLU and HS, respectively. Sig. + RF/kNN stands for model signature with Random Forest/kNN prediction (§ 4.2.2). Mean absolute error (MAE) is the %p difference in accuracy, and Rank is the Spearman rank correlation between the true model ranking and the estimated model ranking.
† Results for Metabench are not directly comparable, as it requires more examples to converge: 150 datapoints for MMLU and ARC (+50%), 450 for HS (+350%), and 200 for WG (+100%). Confidence intervals in App. D.

this concern, we introduce the *chronological split*, where the source models $\mathcal{F}$ consist of models published before January 13, 2024, and the meta test set consists of models after the cutoff date. The train-test ratio is 9:1.

## 5.3 MAIN RESULTS

Table 1 shows the main results. Uniform random sampling, together with direct evaluation with the corresponding annotated labels, yields 3.45%p MAE and .916 rank correlation at 100 samples. The approaches introduced in tinyBenchmarks Polo et al. (2024) improve over this baseline, confirming their findings.

We measure the efficacy of DISCO in two steps: adopt a model-signature approach on top of uniform random sample selections first, and then consider sampling according to predictive diversity scoring (PDS). Even without PDS, on uniform random samples, model signatures are achieving 1.81%p MAE and .933 rank correlation with Random Forest (RF), reaching the state-of-the-art performance with simple and practical ingredients. When PDS is further considered for sample selection, to diversify the model outputs, we achieve 1.07%p MAE and .987 rank correlation (see Appendix C for qualitative comparison of predicted ranks for DISCO vs direct evaluation), demonstrating a significant leap over the prior state of the art from tinyBenchmarks Polo et al. (2024) from ICML 2024.

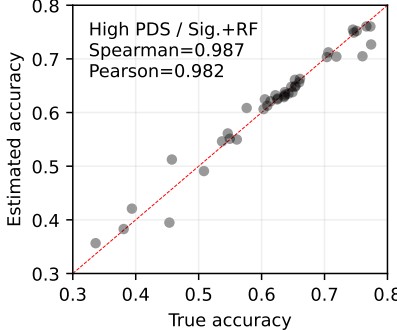

Figure 4: **True and estimated performance on MMLU**. Scatter plot of the performances of 40 models.

To provide an understanding of the distributional comparison of the true model performances and the estimated performances, we show a scatter plot in Figure 4. As signified by the high Spearman's correlation coefficient of .987, the estimated performances closely follow the true performances.

Figure 5 shows the performance against varying degrees of the test set reduction. We observe that the ranking of estimated evaluation methodologies does not change much across a wide range of degrees of reduction. In particular, our DISCO is consistently the best method across all ranges of the number of samples involved. For the extreme rates of compression, at 10 samples, the non-

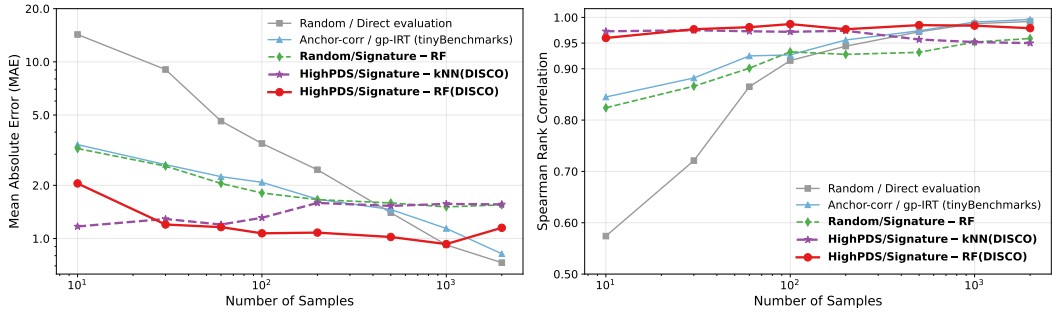

Figure 5: **MMLU performance estimation vs. compression rates**. Mean absolute error (MAE), measured in %p difference in accuracy, and the Spearman rank correlation between the true model ranking and the estimated model ranking are shown. At 100 samples, the results are identical to Table 1. **Main observations**: DISCO hits a better efficiency-precision trade-off across the entire range of compression rates. For an extreme compression rate, kNN is a better choice than random forest (RF).

parametric performance predictor of kNN yields better performance than the parametric Random Forest, suggesting that non-parametric approaches may be more suitable at extreme compression.

## 5.4 FACTOR ANALYSIS

We analyse the impact of several design choices involved in our DISCO on the MMLU dataset. See Table 2 for an overview.

**Model split.** In a recent benchmark for efficient LLM evaluation Zhang et al. (2025), the authors observed that prediction performance drops sharply when test models outperform training models. We extend this idea by replacing performance-based splits with chronological splits, training on older models and testing on newer ones. This better reflects real-world usage, whereas performance-based splits create an artificial stress test.

For this purpose, we introduced the *chronological split* in §5.2. We examine the impact of this model splitting on the result. We observe that our DISCO is robust to the choice of splitting strategy. Chronological splitting yields a rank correlation of .987, which is nearly identical to the .986 obtained with uniform splitting (Table 2 (a)).

**Stratification.** We measure the efficacy of the stratification strategy in (Polo et al., 2024), where equal numbers of anchor points are selected from each of 57 tasks in the MMLU dataset (Table 2 (b)). We find that stratification (.978) is not effective when data points are sampled according to PDS (.987).

**Number of source models.** We analyse the sensitivity of DISCO to the number of source models $|\mathcal{F}|$ (Table 2 (c)). With only 100 models (.969 rank correlation), it already outperforms TinyBenchmarks, which uses all 382 available source models (.927 in Table 1). As the number of source models increases, rank correlation steadily improves, reaching a maximum of .987 for $|\mathcal{F}| = 382$.

**Dimensionality reduction.** We compare PCA with different target dimensions to Uniform Manifold Approximation and Projection (UMAP) (McInnes et al., 2020) for dimensionality reduction (Table 2 (d)). We notice that dimensionality reduction helps reduce potential overfitting: without it (using all 3100 dimensions), the correlation is .918, while with PCA at 256 dimensions, it improves to .987. Overall, PCA outperforms UMAP and remains robust across a wide range of dimensions.

**Prediction model.** We consider a wide range of prediction models (Table 2 (e)). Random Forest achieves the highest rank correlation of .987, outperforming all other methods.

| Split | Rank |
|---|---|
| **Chrono.** | **0.987** |
| Uniform | 0.986 |

(a) Model split

| Strat. | Rank |
|---|---|
| Yes | 0.978 |
| **No** | **0.987** |

(b) Stratification

| $|\mathcal{F}|$ | Rank |
|---|---|
| 3 | 0.652 |
| 10 | 0.787 |
| 30 | 0.844 |
| 100 | 0.969 |
| 300 | 0.975 |
| **382** | **0.987** |

(c) # Source models

| Method | Dims | Rank |
|---|---|---|
| None | 3100 | 0.918 |
| UMAP | 64 | 0.982 |
| UMAP | 128 | 0.970 |
| UMAP | 256 | 0.970 |
| PCA | 32 | 0.983 |
| PCA | 64 | 0.987 |
| PCA | 128 | 0.987 |
| **PCA** | **256** | **0.987** |
| PCA | 300 | 0.985 |

(d) Dim reduction

| Method | Rank |
|---|---|
| **kNN** | **0.971** |
| **Rand. Forest** | **0.987** |
| 2-layer MLP | 0.959 |
| 3-layer MLP | 0.962 |
| Linear reg. | 0.888 |
| Ridge reg. | 0.918 |
| Lasso | 0.908 |
| Grad boosting | 0.980 |

(e) Prediction model

Table 2: **Factor analysis for DISCO on MMLU**. Highlighted in bold are the default design choices for DISCO. All comparisons are based on 100 selected samples.

## 5.5 RESULTS FOR VISION DOMAIN

In this section, we give a quick overview of the DISCO applied to the vision domain. For detailed results, see Appendix J.

**Setup.** We use ImageNet-1k (Russakovsky et al., 2015) with 1.28M images and 400 pre-trained models from `timm` (Wightman, 2019), spanning convolutional (Krizhevsky et al., 2012) and transformer (Dosovitskiy et al., 2021) architectures (0.3M–300M parameters). Following the language domain, we adopt a *chronological split* with a cutoff of 5 April 2023 (88:12 train–test). Performance is evaluated using mean absolute error (MAE) and Spearman's rank correlation. The details on the baselines for the vision domain are in Appendix J.2.

**Results.** Our DISCO approach significantly compresses the ImageNet validation set by reducing it to just 100 data points, achieving an inference cost reduction of 99.8%. DISCO with uniform random sampling and random forest prediction on model signatures achieves 0.86%p MAE and .944 rank correlation, surpassing the baseline. Using a predictive diversity score

| Approach | Selection §4.1 | Prediction §4.2 | MAE↓ | Rank↑ |
|---|---|---|---|---|
| Baseline | Random | Direct eval. | 3.03 | 0.652 |
| Lifelong Bench. | Uniform correctness | Weighted sum | 2.06 | 0.838 |
| SSEPY | Uniform confidence | Weighted sum | 3.05 | 0.762 |
| Model signature | Random | Sig. + kNN | 1.72 | 0.808 |
| | | Sig. + RF | 0.86 | 0.944 |
| DISCO (ours) | High PDS | Sig. + kNN | 1.68 | 0.819 |
| | | Sig. + RF | **0.63** | **0.969** |

Table 3: **DISCO compression of ImageNet validation dataset.** We evaluate the generalisation of our DISCO to the computer vision domain. We reduce the test set to 100 anchor points. The main metrics are mean absolute error (MAE), measured in %p difference in accuracy, and the Spearman rank correlation (Rank) between the true model ranking and the estimated model ranking. **Main observations**: (1) Same as for language experiments, model signature is an effective strategy for performance estimation. (2) Using PDS on top improves performance even more.

(PDS) for data selection and a Random Forest for prediction, our method achieves a 0.63%p MAE and .969 rank correlation, substantially outperforming the baseline (Table 3). The results demonstrate that DISCO is effective in both language and vision domains.

DISCO (.969/0.63) outperforms Lifelong Bench. (Prabhu et al., 2024) (.838/2.06) and SSEPY (Fogliato et al., 2024) (.762/3.05) in both rank correlation and MAE. The conclusion from language experiments holds: instead of selecting anchor points with wide coverage of sample difficulty, one should focus on **selecting the points on which models typically disagree**.

## 6 CONCLUSION

Evaluating ML models is increasingly expensive due to larger models, datasets, and benchmarks. It is especially true for general-purpose LLMs requiring broad evaluation.

We propose DISCO, which selects a small informative subset of the evaluation data and estimates model performance from predictions on it. DISCO cuts evaluation costs by over 99% with minimal error and consistently outperforms prior methods.

This enables practical use: efficient evaluation on limited compute, frequent performance tracking during training, and cheap end-user checks of deployed models.

**Limitations.** The main limitation of DISCO is robustness to distribution shifts in the model population. Shifts can arise from new architectures, training methods, or objectives, which introduce patterns unseen during training and reduce estimator accuracy. Future work could address this with adaptive sample selection or periodic retraining on newer models (see details in Appendix F).

We also discuss unsuitable tasks for DISCO. The main constraint is that DISCO requires predictive probabilities for several predefined answer choices for each question. These answer choices correspond to the classes in Proposition 1 in the original submission. That makes DISCO not suitable for open-ended generation tasks such as translation or summarisation. Applying DISCO to such tasks would first require defining sets of correct and incorrect outputs. We leave such experiments for future work.

## AUTHOR CONTRIBUTIONS

Benjamin, Joon, and Alexander conceived the project. Alexander led the language experiments, Benjamin led the vision experiments. Joon and Martin helped design the experiments. Alexander, Martin, and Joon led the writing of the paper. Martin and Joon provided helpful feedback throughout the project.

## ACKNOWLEDGMENTS

This work was supported by the Tübingen AI Center. AR thanks the International Max Planck Research School for Intelligent Systems (IMPRS-IS) for support. This research utilised compute resources at the Tübingen Machine Learning Cloud, DFG FKZ INST 37/1057-1 FUGG.

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

## DISCLAIMER FOR USE OF LLMS

We primarily used LLMs in coding co-pilot applications to facilitate experimentation and help with plotting code for result presentation. LLMs were also used as writing tools to assist in refining the paper. However, the final version was carefully reviewed and finalized by the authors. No LLMs were used in ideation and experimental design.

## A   EXTENDED EXPERIMENTAL SETUP

**Datasets.** We evaluate DISCO on four widely used language modelling benchmarks:

- Massive Multitask Language Understanding (MMLU) (Hendrycks et al., 2021) question-answering dataset that covers 57 tasks about world knowledge and problem-solving ability.
- HellaSwag (Zellers et al., 2019) dataset that focuses on commonsense natural language inference.
- Winogrande (Sakaguchi et al., 2021): dataset of 273 expert-crafted pronoun resolution problems originally designed to be unsolvable for statistical models.
- AI2 Reasoning Challenge (ARC) (Clark et al., 2018): question-answering dataset that contains only natural, grade-school science questions (authored for human tests) and requires knowledge and reasoning.

## B   COMPUTATIONAL COSTS

We report the space–time complexity for the main stages of DISCO, as well as the cost of direct evaluation of a target model. The numbers correspond to a single H100 GPU and are extrapolated from evaluations of five diverse 32B LLMs on MMLU (standard deviation across 5 runs).

### B.1   DISCO PIPELINE OVERVIEW

The DISCO pipeline consists of two stages: an **offline** stage (run once) and an **online** stage (run for each new target model).

**Offline Stage**

- Evaluate $M$ source models on the full test dataset ($M = 385$ in this experiment)
- Store source model outputs
- Select 100 anchor points that maximise PDS/JSD
- Concatenate outputs on anchor points to form model signatures
- Train a predictor to estimate model performance on the full test dataset from these signatures

**Online Stage**

- Evaluate one target model on the 100 anchor points
- Store target model outputs
- Concatenate to obtain the target model signature
- Run the predictor to estimate performance on the full test dataset

For every target model, the anchor points and predictor trained offline are reused.

### B.2   COST METRICS

The majority of the compute is required by the offline stage (3284 GPU-hours).

| | |
|---|---|
| DISCO: offline stage | $3284.05 \pm 592.90$ GPU-hours |
| DISCO: online stage | $0.07 \pm 0.01$ GPU-hours |
| Direct evaluation | $8.53 \pm 1.54$ GPU-hours |

Table 4: DISCO computation cost metrics (single H100 GPU, MMLU, 5 runs) compared to direct evaluation cost. Computation savings are computed as the difference between the direct evaluation cost and online computation cost, i.e., $8.53 - 0.07 = 8.46$ GPU-hours (mean). This yields $8.46 \pm 1.54$ GPU-hours saved per evaluated model.

| | |
|---|---|
| Source outputs (offline) | 86.54 MB |
| Source signatures (offline) | 400 KB |
| Target outputs (online) | 224.78 KB |
| Target signature (online) | 1 KB |

Table 5: DISCO storage requirements (offline stage for 400 source models and online stage for one target model).

## B.3 BREAK-EVEN ANALYSIS: HOW MANY EVALUATIONS JUSTIFY DISCO SETUP?

DISCO breaks even at **389 evaluations**. Since each DISCO evaluation saves $8.46$ GPU-hours per model (vs. $8.53$ GPU-hours direct evaluation, minus $0.07$ GPU-hours online DISCO cost), the break-even point is:

$$389 = \frac{3284}{8.46}.$$

In practice, hundreds of checkpoint evaluations naturally occur during model development. For example, a single OLMo-2-32B (OLMo et al., 2024) training run includes **753 checkpoints** on Hugging Face, already exceeding break-even.

In some cases, there is no need to evaluate source models at all if offline predictions are downloaded from platforms such as open-llm-leaderboard.

## B.4 COMPARISON TO ALTERNATIVE APPROACHES

We briefly remind the pipelines of the compared methods:

- **Selection**: select a set of anchor points, i.e., a subset of the full test dataset based on different signals (random, IRT, model disagreement, etc.).

- **Prediction**: estimate model performance on the full test dataset from outputs on anchor points.

That is why we use "Selection" and "Prediction" columns to explain the difference between methods. See § 4.1 for details on selection methods, and § 4.2 for details on prediction methods.

| Method | Selection | Prediction | Offline (GPU-h) | Online (GPU-s) |
|---|---|---|---|---|
| Baseline | – (use all samples) | Direct eval | – | $30739 \pm 5514$ |
| Baseline | Random | Direct eval | – | $218 \pm 39$ |
| tinyBenchmarks | Random | gp-IRT | $3284 \pm 592$ | $219 \pm 39$ |
| tinyBenchmarks | Anchor-corr | gp-IRT | $3284 \pm 592$ | $219 \pm 39$ |
| tinyBenchmarks | Anchor-IRT | gp-IRT | $3284 \pm 592$ | $219 \pm 39$ |
| DISCO | High-PDS | RF | $3284 \pm 592$ | $218 \pm 39$ |
| DISCO | High-PDS | KNN | $3284 \pm 592$ | $218 \pm 39$ |

Table 6: Comparison to alternative approaches: offline (GPU-h) and online (GPU-s) costs, and computation savings (GPU-h). DISCO dominates all other methods.

The differences in online cost across methods are negligible (e.g., 219 vs. 218 GPU-seconds). Offline costs are equal up to rounding. Efficient evaluation methods allow for saving $\frac{(30739-218)}{30739} \cdot 100\% = 99.3\%$ of evaluation cost in comparison to full evaluation.

## C  QUALITATIVE MEANING OF RANK CORRELATION IMPROVEMENTS

To justify the additional computation required for DISCO relative to direct evaluation, we illustrate how the increase in rank correlation from 91.6 (direct evaluation) to 98.7 (DISCO) in Table 3 translates into qualitative improvements in model ranking.

Figure 6 includes scatter plots of true vs. predicted ranks (see Figure 6). The direct-evaluation predictor demonstrates noticeable spread around the diagonal, while DISCO's predictions align almost perfectly with it, indicating substantially more reliable ranking.

### Ground-truth vs Predicted Rank

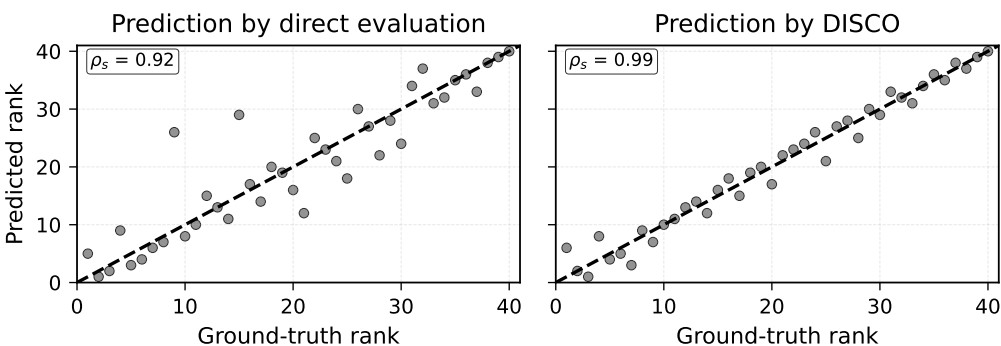

Figure 6: True vs. predicted rank comparison: direct evaluation vs. DISCO. $\rho_s$ means Spearman rank correlation.

## D  ADDITIONAL EVALUATION RESULTS

### D.1  REPORT CONFIDENCE INTERVALS

We report the standard deviation for the previously reported results on MMLU from Table 3, evaluated over one fixed chronological split and 5 independent runs.

We briefly remind the pipelines of the compared methods:

- **Selection**: select a set of anchor points, i.e., a subset of the full test dataset based on different signals (random, IRT, model disagreement, etc.).

- **Prediction**: estimate model performance on the full test dataset from outputs on anchor points.

That is why we use "Selection" and "Prediction" columns to explain the difference between methods. See § 4.1 for details on selection methods, and § 4.2 for details on prediction methods. MAE is mean absolute error; Rank is Spearman's rank correlation.

DISCO results are more stable than those of IRT and random sampling. This is because the only random component is Random Forest initialisation. In contrast, IRT is trained using variational inference, where stochastic gradient optimisation introduces additional randomness beyond model parameter initialisation.

| Method | Selection | Prediction | MAE ↓ | Rank ↑ |
|---|---|---|---|---|
| Baseline | Random | Direct evaluation | $3.45 \pm 0.67$ | $91.6 \pm 2.6$ |
| tinyBenchmarks | Random | gp-IRT | $2.79 \pm 0.20$ | $92.2 \pm 2.3$ |
| tinyBenchmarks | Anchor-corr | gp-IRT | $2.08 \pm 0.20$ | $92.7 \pm 2.1$ |
| tinyBenchmarks | Anchor-IRT | gp-IRT | $3.25 \pm 0.49$ | $92.2 \pm 1.5$ |
| DISCO | High JSD | KNN | $1.14 \pm 0.00$ | $97.5 \pm 0.0$ |
| DISCO | High JSD | RF | $1.30 \pm 0.02$ | $98.7 \pm 0.1$ |
| DISCO | High PDS | KNN | $1.31 \pm 0.00$ | $97.2 \pm 0.0$ |
| DISCO | High PDS | RF | $1.07 \pm 0.04$ | $98.7 \pm 0.2$ |

Table 7: MMLU, single chronological split: MAE (%p) and Spearman rank (%), mean $\pm$ std over runs. DISCO (PDS/JSD + RF/kNN) has lower variance than baselines.

### D.2 MULTIPLE TRAIN/TEST SPLIT FOR CHRONOLOGICAL EVALUATION

To expand the number of chronological splits, we bootstrap 5 different train/test chronological splits using the following protocol: for each run, we split models into 385 old and 40 new based on timestamps, then bootstrap 346 source and 36 test models from these sets. Details on the chronological split can be seen in § 5.2. Results for the new splits can be seen below.

| Method | Selection | Prediction | MAE ↓ | Rank ↑ |
|---|---|---|---|---|
| Baseline | Random | Direct evaluation | $2.85 \pm 0.85$ | $93.3 \pm 3.0$ |
| tinyBenchmarks | Random | gp-IRT | $2.42 \pm 0.43$ | $93.6 \pm 2.5$ |
| tinyBenchmarks | Anchor-corr | gp-IRT | $1.93 \pm 0.31$ | $92.9 \pm 3.0$ |
| tinyBenchmarks | Anchor-IRT | gp-IRT | $3.13 \pm 0.33$ | $90.2 \pm 4.5$ |
| DISCO | High PDS | KNN | $1.23 \pm 0.09$ | $97.0 \pm 1.1$ |
| DISCO | High PDS | RF | $1.25 \pm 0.14$ | $98.0 \pm 0.6$ |

Table 8: MMLU, bootstrapped chronological train/test splits (5 runs): MAE (%p) and Spearman rank correlation (%), mean $\pm$ std. DISCO remains best across methods.

Bootstrapped chronological splits slightly change the mean values (e.g., rank correlation from 98.6 to 98.0 and MAE from 1.06 to 1.25 for DISCO), but they do not alter the superiority of DISCO over other baselines.

## E SENSITIVITY OF DISCO TO MODEL CALIBRATION

To evaluate the sensitivity of DISCO to model calibration, we compared the Expected Calibration Error (ECE) of target models with the Mean Absolute Error (MAE) between their true performance and DISCO-predicted performance on MMLU. We observe a Pearson correlation of $0.49$ between MAE and ECE, indicating that better-calibrated models lead to more accurate performance (lower MAE) predictions by DISCO.

This phenomenon is explained by the information relationship between confidence and correctness. For a perfectly calibrated model, the mapping between prediction confidence and correctness is deterministic and monotonic, resulting in high mutual information. In contrast, for a highly miscalibrated model (e.g., random guessing or uniformly confident but incorrect), prediction confidence becomes statistically independent of correctness, leading to low mutual information. Consequently, the more calibrated a model is, the more predictive its confidence patterns are of its true performance, and therefore the more informative its signature is for DISCO performance prediction.

The corresponding scatter plot is shown in Figure 7.

During this analysis, we observed that two factors are confounded in calibration metrics: (1) overall confidence level, and (2) how well predictive uncertainty is reflected in confidence. To isolate the effect of overall confidence, we compared MAE with mean prediction confidence separately. We find

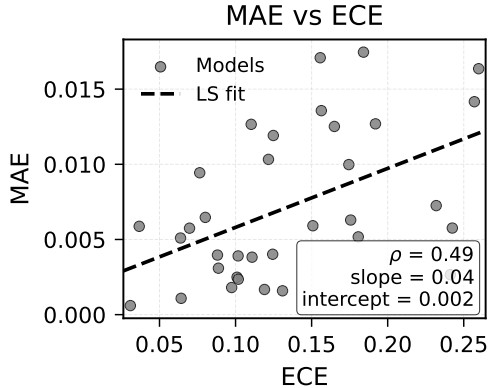

Figure 7: Correlation between DISCO prediction error (MAE) and Expected Calibration Error (ECE).

a Pearson correlation of $-0.47$ between MAE and mean confidence, suggesting that overall model confidence is the dominant component of ECE that influences DISCO performance.

Figure 8 presents the corresponding scatter plot.

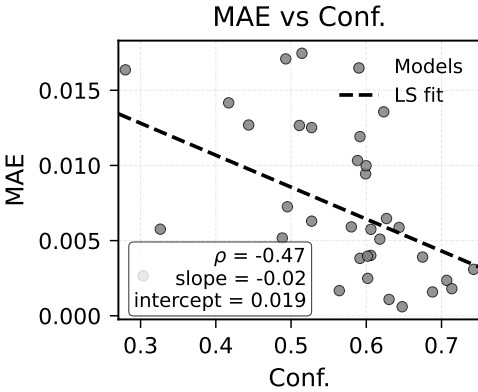

Figure 8: Correlation between DISCO prediction error (MAE) and mean model confidence.

## F    PERFORMANCE GAP EXPERIMENTS

In addition to source/target model splits discussed in § 5.4, we added experiments with a wider performance gap between source and target models to identify potential failure modes for DISCO. Inspired by (Zhang et al., 2025), we introduce a performance split with varying gaps. We sort all models by their average performance and take the top-10% or top-30% (40 or 128 models) as target models, while using the bottom-90% or bottom-50% (385 or 213 models) as source models. The accuracy gap between the weakest target model and the strongest source model is 0.07%p or 8.18%p.

All model splits are summarised in Table 9.

Table 10 reports Spearman's rank correlation.

For a source/target split with a performance gap, the difference between DISCO and direct evaluation is 1.8%p, which is lower than for the IID split (6.5%p), the chronological split (7.1%p), or the performance split without a gap (8.7%p).

|  | IID | Chron. | Performance w/o gap | Perf. w/ gap |
|---|---|---|---|---|
| Prelim. model sorting | – | By timestamp | By performance | By performance |
| Target models | Every 10th model | Top-10% | Top-10% | Top-30% |
| Source models | Everything else | Bottom-90% | Bottom-90% | Bottom-50% |

Table 9: Source/target models splits.

|  | IID | Chron. | Perf. w/o gap | Perf. w/ gap |
|---|---|---|---|---|
| Direct eval on random subset | $92.1 \pm 1.3$ | $91.6 \pm 2.6$ | $89.8 \pm 5.9$ | $87.4 \pm 5.7$ |
| DISCO | $98.6 \pm 0.3$ | $98.7 \pm 0.2$ | $98.1 \pm 0.2$ | $89.2 \pm 1.0$ |
| Mean difference (DISCO – direct eval) | $+6.5$ | $+7.1$ | $+8.7$ | $+1.8$ |

Table 10: DISCO benefit vs direct evaluation on random subset across various source/target models splits.

While this scenario can be seen as a failure mode for DISCO, we believe that such a source/target performance gap does not happen in practice. Instead, the accuracies of source and target models are often mixed. There are two main reasons for that. First, when practitioners develop new models, their early versions are often worse than the best previously evaluated models. Second, it takes time before new models consistently outperform older ones. Infrequent extra evaluations can allow practitioners to always keep the performance gap low. That makes the performance split with a gap less realistic than other splits. **We thus conclude that DISCO does not break when the source and target model distributions differ, but only when the difference is unrealistically substantial.**

## G  MUTUAL INFORMATION AND JENSEN-SHANNON DIVERGENCE

In this section, we show that Mutual Information is equivalent to JSD in our setting. We present the setup and assumptions, then prove the proposition.

### G.0.1  SETUP.

Let $m \sim \mathrm{Unif}\{1, \ldots, M\}$ be the index of a uniformly chosen model. Given $m$, the prediction on datapoint $i$ has categorical law $P_{\hat{Y}_i|m}$. Define the ensemble mean distribution

$$P_{\hat{Y}_i} = \mathbb{E}_{m \sim \mathrm{Unif}[M]}\left[P_{\hat{Y}_i|m}\right] = \frac{1}{M}\sum_{m=1}^{M} P_{\hat{Y}_i|m}.$$

Let $S(m)$ denote any statistic that is a deterministic function of $m$ computed on $\mathcal{D}$ (e.g. accuracy on $\mathcal{D}$).

### G.0.2  ASSUMPTIONS.

**Assumption A1** (Uniform prior). *The model index is uniformly distributed:* $m \sim \mathrm{Unif}\{1, \ldots, M\}$.

We note that Assumption A1 does not assume a uniform prior over the source models $f^1, \ldots, f^M$. It only assumes that the model index is drawn uniformly, $m \sim \mathrm{Unif}1, \ldots, M$. If the prior over models is non-uniform, we can replicate models proportionally to their sampling probabilities. In this case, the index distribution can be made uniform without changing the resulting model sampling outcomes.

**Assumption A2** (Deterministic predictions). *Conditional on $m$, each prediction $\hat{Y}_i$ is fully determined by $m$ (or more generally, any residual randomness is independent across $i$ and independent of $m$).*

### G.0.3  PROPOSITION.

**Proposition 3.** *Under Assumptions A2–A1, if $S(m)$ is injective, then*

$$\mathrm{MI}_{m,\hat{Y}}\left(S(m); \hat{Y}_i\right) = \mathcal{H}_{\hat{Y}_i}\left(P_{\hat{Y}_i}\right) - \mathbb{E}_{m \sim \mathrm{Unif}[M]}\left[\mathcal{H}_{\hat{Y}_i}\left(P_{\hat{Y}_i|m}\right)\right] =: \mathrm{JSD}\left(\{P_{\hat{Y}_i|m}\}_{m=1}^{M}\right).$$

*Proof.* By Assumption A2 and since $S(m)$ is a deterministic function of $m$, we have the Markov chain

$$\widehat{Y}_i \longleftrightarrow m \longleftrightarrow S(m).$$

If $S$ is injective, then $m$ is recoverable from $S(m)$, hence

$$I\big(S(m); \widehat{Y}_i\big) = I\big(m; \widehat{Y}_i\big).$$

By the definition of mutual information,

$$I(m; \widehat{Y}_i) = \mathcal{H}_{\widehat{Y}_i}\Big(P_{\widehat{Y}_i}\Big) - \mathbb{E}_m\Big[\mathcal{H}_{\widehat{Y}_i}\Big(P_{\widehat{Y}_i|m}\Big)\Big].$$

*Marginal distribution (using Assumption A1):*

$$P_{\widehat{Y}_i} = \sum_{m=1}^{M} \Pr(m) \, P_{\widehat{Y}_i|m} = \tfrac{1}{M} \sum_{m=1}^{M} P_{\widehat{Y}_i|m}.$$

Thus $\mathcal{H}_{\widehat{Y}_i}(P_{\widehat{Y}_i})$ is the entropy of the mixture distribution.

*Conditional entropy (using Assumption A1):*

$$\mathbb{E}_m\big[\mathcal{H}_{\widehat{Y}_i}(P_{\widehat{Y}_i|m})\big] = \tfrac{1}{M} \sum_{m=1}^{M} \mathcal{H}_{\widehat{Y}_i}(P_{\widehat{Y}_i|m}).$$

*Combine:*

$$I(m; \widehat{Y}_i) = \mathcal{H}_{\widehat{Y}_i}(P_{\widehat{Y}_i}) - \tfrac{1}{M} \sum_{m=1}^{M} \mathcal{H}_{\widehat{Y}_i}(P_{\widehat{Y}_i|m}) =: \mathrm{JSD}\big(\{P_{\widehat{Y}_i|m}\}_{m=1}^{M}\big).$$

$\square$

We note that JSD and, as a consequence, DISCO predictor directly depend on the heterogeneity of source models. This heterogeneity is captured by how distinguishable the model-conditional predictive distributions $P_{\widehat{Y}_i|m}$ are from their mixture, as measured by their average KL divergence to the ensemble mean. The larger this KL divergence, the higher the JSD. According to Proposition 3, a larger JSD is equivalent to higher mutual information between outputs and benchmark accuracy, which leads to better performance of the DISCO predictor. Conversely, if this KL divergence is small, the JSD is also small. In particular, if we have many copies of the same model, then JSD as well as mutual information become zero, leading to poor predictor performance.

## H  BOUNDS FOR JENSEN-SHANNON DIVERGENCE (JSD) VIA PREDICTIVE DIVERSITY SCORE (PDS)

In this section, we show that JSD is bounded quadratically below and linearly above by PDS. We first relate JSD to total variation (§ H.1), then show total variation is monotone in PDS (§ H.2), and then combine these results in § H.3.

### H.1  BOUNDS FOR JSD VIA TOTAL VARIATION (TV)

We begin by showing that JSD is bounded quadratically below and linearly above by total variation. We first introduce the setup with required definitions (§ H.1.1), then prove the proposition (§ H.1.2).

### H.1.1  SETUP.

Let $\{P_{\widehat{Y}_i|m}\}_{m=1}^{M}$ be distributions on $K$ classes. Define the mixture

$$\bar{P} = \frac{1}{M} \sum_{m=1}^{M} P_{\widehat{Y}_i|m}.$$

**Definition 1** (Jensen–Shannon divergence).

$$\mathrm{JSD}\big(\{P_{\widehat{Y}_i|m}\}\big) = \frac{1}{M}\sum_{m=1}^{M} D_{\mathrm{KL}}(P_{\widehat{Y}_i|m}\|\bar{P}) = \mathcal{H}(\bar{P}) - \tfrac{1}{M}\sum_{m=1}^{M}\mathcal{H}(P_{\widehat{Y}_i|m}).$$

**Definition 2** (Total variation). *For distributions $P, Q$ on the same support,*

$$\mathrm{TV}(P,Q) = \tfrac{1}{2}\|P - Q\|_1.$$

### H.1.2    PROPOSITION.

Now, we show that JSD is bounded quadratically below and linearly above by total variation.

**Proposition 4** (JSD–TV sandwich bounds). *For any $M \geqslant 2$ distributions $\{P_{\widehat{Y}_i|m}\}_{m=1}^M$ with mixture $\bar{P}$,*

$$\frac{2}{\ln 2}\cdot\frac{1}{M}\sum_{m=1}^{M}\mathrm{TV}(P_{\widehat{Y}_i|m},\bar{P})^2 \;\leqslant\; \mathrm{JSD}\big(\{P_{\widehat{Y}_i|m}\}_{m=1}^M\big) \;\leqslant\; \frac{M}{M-1}\log M\cdot\frac{1}{M}\sum_{m=1}^{M}\mathrm{TV}(P_{\widehat{Y}_i|m},\bar{P}).$$

*Proof. Lower bound.* By Pinsker's inequality (e.g. Equation 1 in (Sason, 2015)),

$$D_{\mathrm{KL}}(P\|Q) \;\geqslant\; \frac{2}{\ln 2}\,\mathrm{TV}(P,Q)^2.$$

Substituting $Q = \bar{P}$ and averaging over $m$ yields the lower bound.

*Upper bound.* Fix $m$. Write

$$\bar{P} = \alpha P_{\widehat{Y}_i|m} + (1-\alpha)\zeta, \quad \alpha = \tfrac{1}{M}, \quad \zeta = \tfrac{1}{M-1}\sum_{s\neq m} P_{\widehat{Y}_i|s}.$$

Define $t(i) = \zeta(i)/P_{\widehat{Y}_i|m}(i)$ when $P_{\widehat{Y}_i|m}(i) > 0$ (set $t(i) = +\infty$ if $P_{\widehat{Y}_i|m}(i) = 0, \zeta(i) > 0$). Then $\mathbb{E}_{P_{\widehat{Y}_i|m}}[t] = 1$ and

$$D_{\mathrm{KL}}(P_{\widehat{Y}_i|m}\|\bar{P}) = \mathbb{E}_{P_{\widehat{Y}_i|m}}\big[-\log\big(\alpha + (1-\alpha)t\big)\big].$$

Let $f(u) = -\log(\alpha + (1-\alpha)u)$, $u \geqslant 0$. Then $f$ is convex, decreasing, with $f(1) = 0$, $f(0) = \log(1/\alpha) = \log M$. By convexity,

$$f(u) \;\leqslant\; (1-u)f(0) = (1-u)\log M.$$

Thus,

$$D_{\mathrm{KL}}(P_{\widehat{Y}_i|m}\|\bar{P}) \leqslant \log M\cdot\mathbb{E}_{P_{\widehat{Y}_i|m}}\big[(1-t)\big] \leqslant \log M\cdot\mathbb{E}_{P_{\widehat{Y}_i|m}}\big[(1-t)_+\big].$$

Now

$$\mathbb{E}_{P_{\widehat{Y}_i|m}}[(1-t)_+] = \sum_i P_{\widehat{Y}_i|m}(i)\max\{0, 1 - \zeta(i)/P_{\widehat{Y}_i|m}(i)\} = \sum_i (P_{\widehat{Y}_i|m}(i) - \zeta(i))_+.$$

By the balance-of-deviations identity (§ 1),

$$\sum_i (P_{\widehat{Y}_i|m}(i) - \zeta(i))_+ = \mathrm{TV}(P_{\widehat{Y}_i|m},\zeta).$$

Finally, since $\bar{P} = \alpha P_{\widehat{Y}_i|m} + (1-\alpha)\zeta$, one has

$$\mathrm{TV}(P_{\widehat{Y}_i|m},\zeta) = \tfrac{M}{M-1}\,\mathrm{TV}(P_{\widehat{Y}_i|m},\bar{P}).$$

Combining yields

$$D_{\mathrm{KL}}(P_{\widehat{Y}_i|m}\|\bar{P}) \;\leqslant\; \tfrac{M}{M-1}\,\log M\cdot\mathrm{TV}(P_{\widehat{Y}_i|m},\bar{P}).$$

Averaging over $m$ gives the upper bound. $\qquad\square$

**Remark 1.** *The lower bound is quadratic in total variation, the upper bound linear. Thus, JSD interpolates between quadratic growth near equality and linear growth in worst-case separation.*

## H.2 BOUNDS FOR TOTAL VARIATION VIA PREDICTIVE DIVERSITY SCORE

We next show that total variation is monotone in PDS. We introduce the setup with definitions and lemmas (§ H.2.1), then prove the proposition (§ H.2.2).

### H.2.1 SETUP.

Fix a class $c$. Let $X_m = P_{\hat{Y}_i|m}(c)$ and $\mu = \bar{P}(c)$. Define:

**Definition 3** (Envelope and spread, per class).

$$E_c = \max_m X_m - \mu, \qquad U_c = \frac{1}{2M} \sum_{m=1}^{M} |X_m - \mu|.$$

**Definition 4** (Predictive Diversity Score).

$$\mathrm{PDS}\big(\{P_{\hat{Y}_i|m}\}\big) = \sum_{c=1}^{K} \max_m P_{\hat{Y}_i|m}(c).$$

**Lemma 1** (Balance-of-deviations identity). *For any $a_1, \ldots, a_M$ with $\sum_m a_m = 0$, writing $a_+ = \max\{0, a\}$,*

$$\sum_{m=1}^{M} (a_m)_+ = \sum_{m=1}^{M} (a_m)_- = \tfrac{1}{2} \sum_{m=1}^{M} |a_m|.$$

*Proof.* Decompose $a = a_+ - a_-$, $|a| = a_+ + a_-$. Summing and using $\sum_m a_m = 0$ gives

$$\sum_m a_{m,+} - \sum_m a_{m,-} = 0 \quad \Rightarrow \quad \sum_m a_{m,+} = \sum_m a_{m,-}.$$

Then

$$\sum_m |a_m| = \sum_m (a_{m,+} + a_{m,-}) = 2 \sum_m a_{m,+}.$$

$\square$

Applying Lemma 1 with $a_m = X_m - \mu$ yields

$$U_c = \tfrac{1}{M} \sum_{m:X_m > \mu} (X_m - \mu).$$

### H.2.2 PROPOSITION.

Now, we show that total variation is monotone in PDS.

**Proposition 5** (Spread–envelope bounds). *Use notation from Appendix H.2.1. For each class $c$, if at most $z$ models satisfy $X_m > \mu$, then*

$$\frac{1}{M} E_c \leqslant U_c \leqslant \frac{z}{M} E_c.$$

*Aggregating over classes,*

$$\frac{1}{M} E \leqslant U \leqslant \frac{z}{M} E,$$

*where*

$$E = \sum_{c=1}^{K} E_c, \qquad U = \sum_{c=1}^{K} U_c = \tfrac{1}{M} \sum_{m=1}^{M} \mathrm{TV}(P_{\hat{Y}_i|m}, \bar{P}).$$

*Proof.* If $E_c = 0$, then $X_m = \mu$ for all $m$ so $U_c = 0$. Otherwise, let $m^\star = \arg\max_m X_m$. Then

$$U_c = \tfrac{1}{M} \sum_{m:X_m > \mu} (X_m - \mu) \geqslant \tfrac{1}{M}(X_{m^\star} - \mu) = \tfrac{1}{M} E_c.$$

For the upper bound, each positive term is at most $E_c$, and there are at most $z$ such terms, hence

$$U_c \leqslant \tfrac{z}{M} E_c.$$

Summing over classes gives the aggregated bound. $\square$

### H.3 FINAL SANDWICH INEQUALITY

Finally, we combine results from § H.1.1 and § H.2 to show that JSD is bounded quadratically below and linearly above by PDS.

**Proposition 6** (JSD–PDS sandwich). *Use notation from Proposition 1.*

$$\frac{2}{M^2 \ln 2} \left( \text{PDS}\big(\{\text{P}_{\hat{Y}_i|m}\}\big) - 1\right)^2 \;\leqslant\; \text{JSD}\big(\{P_{\hat{Y}_i|m}\}\big) \;\leqslant\; \frac{M}{M-1} \log M \cdot \left(\text{PDS}\big(\{\text{P}_{\hat{Y}_i|m}\}\big) - 1\right).$$

*Proof.* From Theorem 4,

$$\text{JSD} \;\geqslant\; \tfrac{2}{\ln 2} U^2, \qquad \text{JSD} \;\leqslant\; \tfrac{M}{M-1} \log M \cdot U.$$

Define

$$E := \sum_{c=1}^{K} E_c, \qquad U := \sum_{c=1}^{K} U_c.$$

By the definitions,

$$U \;=\; \sum_{c=1}^{K} \frac{1}{2M} \sum_{m=1}^{M} |P_{\hat{Y}_i|m}(c) - \bar{P}(c)| := \frac{1}{2M} \sum_{m=1}^{M} \|P_{\hat{Y}_i|m} - \bar{P}\|_1 := \frac{1}{M} \sum_{m=1}^{M} \text{TV}(P_{\hat{Y}_i|m}, \bar{P}),$$

where $\bar{P} = P_{\hat{Y}_i} = \frac{1}{M} \sum_{m=1}^{M} P_{\hat{Y}_i|m}$, and

$$E \;=\; \sum_{c=1}^{K} \left( \max_m P_{\hat{Y}_i|m}(c) - \bar{P}(c) \right) = \text{PDS} - 1.$$

From Proposition 5,

$$\tfrac{1}{M}(\text{PDS} - 1) \leqslant U \leqslant \tfrac{z}{M}(\text{PDS} - 1).$$

Combining and noticing that $1 \leqslant z \leqslant M$ yields the quadratic lower bound and linear upper bound in $(\text{PDS} - 1)$. $\qquad\square$

## I IMPLEMENTATION DETAILS

The results of all experiments were averaged over five runs. For the language tasks, we either used pre-computed LLM outputs downloaded from the open-llm-leaderboard (Aidar Myrzakhan, 2024) on Hugging Face or computed the outputs using lm-evaluation-harness (Gao et al., 2024). LLM evaluation was performed on a single NVIDIA H100 GPU, while parametric prediction methods (described in § 4.2.2) were trained on a single NVIDIA GTX 2080 Ti GPU. For the vision tasks, we used models from the `timm` library (Wightman, 2019), also evaluated on a single NVIDIA GTX 2080 Ti GPU.

Training a parametric model takes less than one minute for both vision and language domains. Details on LLM evaluation time can be seen in Table 4.

For the MMLU dataset, we observed that computing disagreement scores (as described in § 4.1.2) using all available source models led to worse DISCO performance than using only a subset of them. This can be explained by the fact that including additional, highly similar or redundant models may dilute the effective heterogeneity of the ensemble, which is crucial for DISCO as discussed in § G.0.3. Consequently, we select subsets of source models when computing disagreement scores. These subsets are obtained by randomly sampling $M$ models from the source models, where $M$ is treated as a hyperparameter and tuned jointly with other hyperparameters. For MMLU, the selected value was $M = 100$. We chose random sampling here as it provides a simple and unbiased way to control ensemble size without introducing additional selection criteria.

Details for prediction methods:

- **kNN**: We used $k = 1$ in all experiments unless stated otherwise.

- **Random Forest (RF)**: Implemented using scikit-learn (Pedregosa et al., 2011) with default parameters.
- **2-Layer MLP**: Trained for 200 epochs using the AdamW optimiser (Loshchilov & Hutter, 2019) with default settings and a learning rate of 0.001. Hidden dimension: [128].
- **3-Layer MLP**: Trained for 700 epochs using the AdamW optimiser (Loshchilov & Hutter, 2019) with default settings and a learning rate of 0.001. Hidden dimensions: [128, 128].
- **Linear Regression**: Implemented using scikit-learn (Pedregosa et al., 2011) with default parameters.
- **Ridge Regression**: Implemented using scikit-learn (Pedregosa et al., 2011), with default parameters and regularisation weight $\lambda = 10$.
- **Lasso Regression**: Implemented using scikit-learn (Pedregosa et al., 2011), with default parameters and regularisation weight $\lambda = 0.0001$.
- **Gradient Boosting (GB)**: Implemented using scikit-learn (Pedregosa et al., 2011), with default parameters and 200 base estimators.

Details for dimensionality reduction methods:

- **PCA**: We used the scikit-learn implementation (Pedregosa et al., 2011) and varied the number of principal components as shown in Table 2.
- **UMAP**: We used the `umap-learn` library (McInnes et al., 2018) and varied the number of components as specified in Table 2.

## J  VISION RESULTS

We introduce the setup (§J.1), describe baselines (§J.2), and present results (§J.3).

### J.1  SETUP

**Dataset.** We use ImageNet-1k (Russakovsky et al., 2015) with 1.28 million images. **Models.** We consider 400 models from `timm` (Wightman, 2019) that are pretrained on ImageNet. The models cover convolutional (Krizhevsky et al., 2012) and transformer (Dosovitskiy et al., 2021) architectures. Model sizes range from 0.3M to 300M parameters.

**Model Split.** As in the language domain (§ 5.2), we use the *chronological split*. The cutoff date is 5 April 2023. The train-test ratio of models is 88:12.

**Metrics.** We use mean absolute error (MAE) and Spearman's rank correlation between the true and predicted performances.

**Evaluation.** Evaluation protocol follows the one in § 5.1

### J.2  ABOUT BASELINES FOR VISION DOMAIN

Our work is not the first to propose efficient evaluation in the vision domain. The two closest methods are Lifelong Benchmark (Prabhu et al., 2024) (NeurIPS 2024) and SSEPY (Fogliato et al., 2024) (ECCV 2024). They propose efficient evaluation methods for visual models, using a similar two-stage framework for efficient evaluation in the language domain (see § 4 / Figure 2 of our submission):

1. Select "important/representative" anchor points.
2. Estimate model performance based on model outputs on the anchor points.

In (Prabhu et al., 2024), mean correctness scores across source models are used to measure sample difficulty, and anchor points are selected by sampling every k-th datapoint after sorting them by difficulty (where $k = \frac{\#\text{all datapoints}}{\#\text{anchor points}}$). Final performance is predicted as a weighted sum of correctness scores predicted for each test datapoint. Predicted correctness scores are binary values indicating relative position (after sorting by difficulty) to the hardest anchor point the target model got right. In

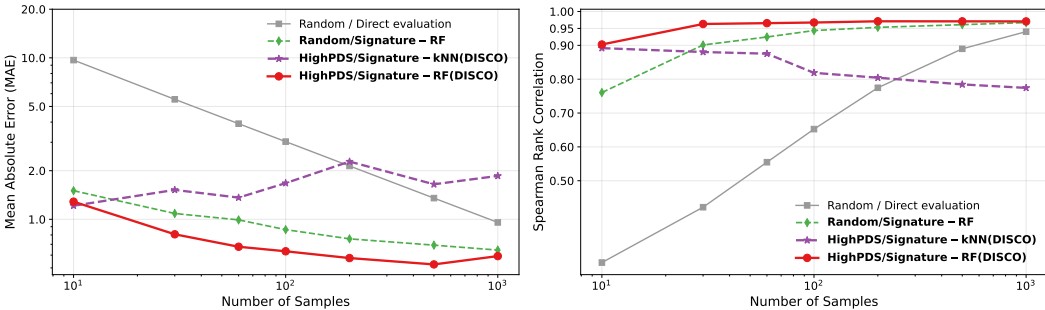

Figure 10: **ImageNet performance estimation vs. compression rates**. Mean absolute error (MAE), measured in %p difference in accuracy, and the Spearman rank correlation between the true model ranking and the estimated model ranking are shown. At 100 samples, the results are identical to Table 3. **Main observations**: Same as for language experiments DISCO hits a better efficiency-precision trade-off across the entire range of compression rates.

(Fogliato et al., 2024), confidence scores of the target model are used to measure sample difficulty. Then samples are clustered by difficulty with K-Means, and anchor points are selected as the centroids of the clusters. Final performance is predicted as a weighted sum of anchor correctness scores. Weights for the weighted sum are determined based on the corresponding cluster sizes using the Horvitz-Thompson estimator.

The main message of our paper is that for selecting anchor points for efficient evaluation, it is better to select **diversity-inducing data points** (DISCO) than to **make a good coverage of sample difficulty** (prior work). In § 5.3, we have shown that our approach beats prior approaches in the language domain.

Likewise, in the vision domain, the existing approaches (Prabhu et al., 2024; Fogliato et al., 2024) focus on a good coverage of sample difficulty rather than on maximizing per-sample information by seeking diversity-inducing data points. We test empirically in Table 3 whether the same conclusion holds in the vision domain by comparing DISCO to (Prabhu et al., 2024; Fogliato et al., 2024).

We computed the results for these baselines ourselves, as the papers do not contain results on ImageNet. For fair comparison, we use the same setup as described in § J.1.

## J.3 MAIN RESULTS

Table 3 shows the main results. See § 5.5 for their overview.

We evaluate the effectiveness of DISCO in two stages. First, we apply the model-signature approach using uniform random sampling. Then, we enhance it by selecting samples based on predictive diversity score (PDS). The results follow a similar trend to the language domain. With uniform random sampling, model signatures combined with Random Forest achieve 0.86%p MAE and a rank correlation of .944, significantly outperforming the naive baseline. Incorporating PDS further improves performance, reaching 0.63%p MAE and a rank correlation of .969.

To illustrate how well the estimated performances align with the true values, we present a scatter plot in Figure 9. The high Pearson correlation coefficient of .970 indicates a strong agreement between the two.

Figure 10 shows performance across varying levels of test set reduction. The relative ranking of evaluation methods remains largely stable, except for the kNN predictor, which degrades as the number of anchor points increases. Notably, DISCO consistently outperforms all baselines, even under extreme compression with as few as 10 samples.

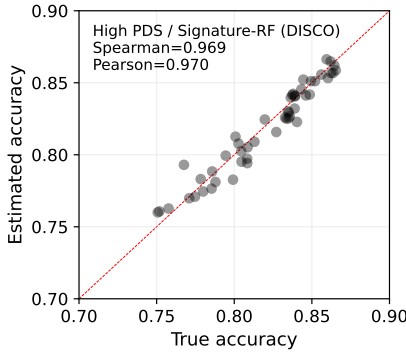

Figure 9: **True and estimated accuracy on ImageNet** for 50 models.

