# OpenReview forum: "DISCO: Diversifying Sample Condensation for Efficient Model Evaluation"
_ICLR.cc/2026/Conference — ICLR 2026 Poster_

### Official Review · Reviewer_zKXy · 2025-10-30

**Soundness:** 3
**Presentation:** 3
**Contribution:** 3
**Rating:** 6
**Confidence:** 4

**Summary:**

The paper proposes DISCO, an efficient evaluation framework that (i) selects a tiny subset of test items by ranking samples via model disagreement (measured with JSD or the Predictive Diversity Score, PDS), and (ii) predicts a target model’s full-benchmark performance from its model signature (concatenated outputs on the selected subset) using simple predictors (kNN or Random Forest). The key thesis is that we should focus on diversifying model responses, rather than on the representativeness of samples, because disagreement maximizes the information about model performance. A formal proposition links mutual information about a model’s score to the Jensen–Shannon divergence across model predictions, motivating the selection rule; DISCO then shows strong empirical results on MMLU, HellaSwag, Winogrande, ARC, and also on ImageNet, often compressing datasets to 100 samples while preserving rank/accuracy with low error. This allows model evaluation with ~99% less inference costs.

**Strengths:**

- Clear practical value: Achieving 99%+ cost reduction while maintaining prediction accuracy addresses a real problem in modern ML evaluation where benchmarks require thousands of GPU hours.
- Elegant simplicity: The approach is conceptually simpler than prior work—avoiding complex clustering algorithms and IRT parameter estimation in favor of greedy sample selection and direct prediction from model signatures.
- Solid empirical validation: Comprehensive experiments across multiple benchmarks (MMLU, HellaSwag, Winogrande, ARC, ImageNet) with hundreds of models, demonstrating consistent improvements over baselines.
- Theoretical motivation: Proposition 1 provides an information-theoretic justification linking mutual information to JSD, showing why model disagreement is informative for performance prediction.
- Domain agnostic: Demonstrates effectiveness in both language (LLMs) and vision (ImageNet) domains.
- Thorough ablations: Table 2 systematically analyzes design choices (model split, stratification, number of source models, dimensionality reduction, prediction model).
- Clarity: The paper is well written, the claims are clear and the experiments support these claims adequately.

**Weaknesses:**

- The empirical results sections are somewhat opaque. I find it hard to understand exactly how the experiments were conducted from the text alone. The experimental set-up seems valid, but perhaps a reworking of the text or an additional, more detailed, description of the empirical set-up could be added in the appendix.
- Statistical rigor: No confidence intervals or significance tests on reported metrics. Single train/test split for chronological evaluation. Unclear if the improvements are statistically significant.
- Limited failure analysis: Insufficient discussion of when/why DISCO might fail. No analysis of which types of benchmarks or tasks are unsuitable. The acknowledged limitation about distribution shift in model populations lacks empirical investigation (see questions, the performance split would be useful)
- Missing cost analysis: No discussion of computational cost for computing PDS scores over full dataset. Cost of training metamodel not reported. How does total cost (sample selection + prediction) compare to just running models on more samples?
- From my understanding the current DISCO pipeline relies on the signature being of the same size (i.e. same size of model outputs and same number of samples), the predictor is then trained on this fixed size representation (that is reduced with PCA). Currently, if one wants to change the number of samples used it would require the entire re-training of the predictor, the main reason seems to be the concatenation of the predictions which treats each scalar prediction *of each sample* as a different variable. I would be very interested in seeing a discussion on how this could be changed so that the pipeline is adaptable with respect to the number of samples (i.e. being able to use a predictor trained on N samples with less or more samples).

**Questions:**

- Missing sentence in the abstract? Before "The typical approach follows two steps." There must be something about efficient sampling (or in that sentence itself) otherwise it's unclear what we're talking about.
- The two assumptions for Proposition 1 should be added in the main text so that result is complete, I believe there is enough space in the paper for this.
- Typo?: line 288 "the stolen"
- Table 2: The PCA results are missing (supposed to be in subplot (d)) while the Prediction model ablation is said to be in (e) while actually being in (d)
- In section 5.3 it is stated that "performance-based splits create an artificial stress test" and that the chronological split better reflects real-world usage. While I generally agree with this statement, I still think a performance-based split should be included for comparison since this is a known failure mode of efficient LLM evaluation methods.
- Why are there no other baselines besides random selection with direct eval. in Table 3 for vision models? Presumably there is nothing preventing past benchmarks to being adapted to this setting.

# Summary
I think this is a strong paper, well written and well justified both theoretically and relative to past works and with adequate empirical evaluations. I would be happy to increase my score if the authors address the identified weaknesses and points in the questions section. Most of these represent relatively small, although important, additions / modifications.

---

> ### Author Response · Authors · 2025-11-21
> **Thank you + W1**
>
> Thank you for your thoughtful review. We appreciate your time and the valuable feedback on experimental clarity, statistical rigor, limitations, and cost analysis.
>
> ## W1. Clarify the experimental setup
>
> We agree that the experimental setup can benefit from clearer exposition. We have updated it. Please see blue text in §5.1 and §A in the revised manuscript.

---

> ### Author Response · Authors · 2025-11-21
> **Response to W2.1 and W2.2**
>
> ## W2.1 Report confidence intervals
> Thank you for the suggestion. We report the standard deviation for the previously reported results on MMLU from Table 1 in the submission on one fixed chronological split over 5 independent runs. We will also update them in the revised manuscript during the next week.
> We briefly remind the pipeline of the compared methods:
> - Selection: select a subset of the full test dataset based on different signals (IRT, model disagreement, etc.) from outputs of source models.
> - Prediction: estimate the target model's performance on the full test dataset from outputs on anchor points.
> We also clarify the table structure: different rows correspond to different methods; columns “Selection”/“Prediction” describe the main pipeline stages; MAE is mean absolute error; Rank is Spearman rank correlation.
>
> | Approach         | Selection    | Prediction       | MAE ↓          | Rank ↑         |
> |------------------|--------------|------------------|----------------|----------------|
> | Baseline         | Random       | Direct evaluation| 3.45 ± 0.67     | 91.6 ± 2.6     |
> | tinyBenchmarks   | Random       | gp-IRT           | 2.79 ± 0.20     | 92.2 ± 2.3     |
> | tinyBenchmarks   | Anchor-corr  | gp-IRT           | 2.08 ± 0.20     | 92.7 ± 2.1     |
> | tinyBenchmarks   | Anchor-IRT   | gp-IRT           | 3.25 ± 0.49     | 92.2 ± 1.5     |
> | DISCO            | High JSD     | KNN              | 1.14 ± 0.00     | 97.5 ± 0.0     |
> | DISCO            | High JSD     | RF               | 1.30 ± 0.02     | 98.7 ± 0.1     |
> | DISCO            | High PDS     | KNN              | 1.31 ± 0.00     | 97.2 ± 0.0     |
> | DISCO            | High PDS     | RF               | 1.07 ± 0.04     | 98.7 ± 0.2     |
>
> DISCO results are more stable than those of IRT and random sampling. This is because the only random component is Random Forest initialisation for RF, while KNN is deterministic. In contrast, IRT is trained using variational inference, where stochastic gradient optimization introduces additional randomness beyond model parameter initialisation.
>
> ## W2.2 Single train/test split for chronological evaluation
>
> To expand the number of chronological splits into source/target models, we bootstrap 5 different train/test chronological splits using the following protocol: for each run, we split models into 385 oldest and 40 newest based on timestamps, then uniformly subsample (bootstrap) 346 source and 36 test models from these sets, respectively. Results for the new splits can be seen below. We will also add it to the revised manuscript during the next week. The table structure is the same as in W2.1.
>
> | Approach         | Selection    | Prediction       | MAE ↓          | Rank ↑         |
> |------------------|--------------|------------------|----------------|----------------|
> | Baseline         | Random       | Direct evaluation| 2.85 ± 0.85     | 93.3 ± 3.0     |
> | tinyBenchmarks   | Random       | gp-IRT           | 2.42 ± 0.43     | 93.6 ± 2.5     |
> | tinyBenchmarks   | Anchor-corr  | gp-IRT           | 1.93 ± 0.31     | 92.9 ± 3.0     |
> | tinyBenchmarks   | Anchor-IRT   | gp-IRT           | 3.13 ± 0.33     | 90.2 ± 4.5     |
> | DISCO            | High PDS     | KNN              | 1.23 ± 0.09     | 97.0 ± 1.1     |
> | DISCO            | High PDS     | RF               | 1.25 ± 0.14     | 98.0 ± 0.6     |
>
> Bootstrapped chronological splits slightly change the mean values (e.g., rank correlation from 98.6 to 98.0 and MAE from 1.06 to 1.25 for DISCO), but they do not alter the superiority of DISCO over other baselines.

---

> ### Author Response · Authors · 2025-11-21
> **Response to W3 + Q5**
>
> ## W3 + Q5. Failure analysis.
> We empirically search for the scenarios where the distribution shift between source and target models causes DISCO to fail. Accordingly, for the MMLU dataset, we introduce a hard split where source models underperform target models. We sort all models by their average performance and take the top 40 models as target models, while using others as source models.
>
> The selection + prediction pipeline of the compared methods and table structure are the same as in W1.
>
> | Approach         | Selection    | Prediction       | MAE ↓          | Rank ↑         |
> |------------------|--------------|------------------|----------------|----------------|
> | Baseline         | Random       | Direct evaluation| 3.21 ± 0.56     | 89.8 ± 5.9     |
> | tinyBenchmarks   | Random       | gp-IRT           | 1.95 ± 0.38     | 92.1 ± 2.9     |
> | tinyBenchmarks   | Anchor-corr  | gp-IRT           | 1.52 ± 0.16     | 95.1 ± 0.5     |
> | tinyBenchmarks   | Anchor-IRT   | gp-IRT           | 2.90 ± 0.30     | 88.0 ± 3.3     |
> | DISCO            | High PDS     | KNN              | 1.49 ± 0.00     | 97.8 ± 0.0     |
> | DISCO            | High PDS     | RF               | 1.30 ± 0.03     | 98.1 ± 0.2     |
>
> In this scenario, DISCO achieves the best MAE (1.30 vs 1.52) and rank correlation (98.1 vs 95.1). Since DISCO outperforms other methods here, this experiment does not cover scenarios where DISCO fails, and we need to extend our empirical failure analysis. Over the next week, we will add experiments with stronger distribution gaps between source and target models to identify potential failure modes for DISCO. We will also include a detailed discussion of unsuitable tasks in the revised manuscript.

---

> ### Author Response · Authors · 2025-11-21
> **Response to W4**
>
> ## W4. Discussion on computational costs
>
> Thank you for the suggestion. We report costs for computing PDS scores over full dataset, cost of training a metamodel, and compare full DISCO cost (sampling + prediction) to direct evaluation on more samples. The numbers are computed for a single H100 GPU and are extrapolated from the evaluation of 5 diverse 32B LLMs on MMLU. Standard deviations are computed across 5 runs.
>
> First, we remind the DISCO pipeline, which consists of two stages: an offline stage run once to set up DISCO, and an online stage run whenever a new target model is evaluated.
>
> - Offline
>   - Evaluate M source models on the full test dataset (M=385 in our experiments)
>   - Store source models' outputs from the previous step
>   - Select 100 anchor points that maximise PDS/JSD
>   - Concatenate outputs on anchor points to get model signatures
>   - Train a predictor that estimates model performance on the full test dataset from model signatures
> - Online
>   - Evaluate 1 target model on 100 anchor points
>   - Store target model output from the previous step
>   - Concatenate outputs to get the target model signature
>   - Run the predictor to estimate the target model’s performance on the full test dataset
>
> For each target model, we reuse the same anchor points and the predictor obtained in the offline stage.
>
> | Item                                                        | Value                          |
> |-------------------------------------------------------------|--------------------------------|
> | Eval source models + compute PDS across dataset (offline)   | 3284.05 ± 593.53 GPU-hours     |
> | Cost of training the metamodel (offline)                    | 17 ± 6 CPU-seconds             |
> | Eval target model on 100 anchor points + apply DISCO (online)| 218 ± 39 GPU-seconds           |
> | Full DISCO: sample selection + prediction                    | 3284.11 ± 593.53 GPU-hours     |
> | Direct evaluation on more samples                           | 8.5 ± 1.5 GPU-hours             |
>
> Most of full DISCO’s offline + online runtime comes from evaluating source models to compute PDS scores (3284.05 GPU-hours). In contrast, the online stage of evaluating the target model with DISCO is significantly faster than a full direct evaluation (218 GPU-seconds vs. 8.5 GPU-hours).
>
> To find out how many DISCO-evaluations of target models justify the offline costs, we report space-time complexity for each step as well as the cost of direct evaluation of the target model.
>
> | Item                                 | Value                           |
> |--------------------------------------|---------------------------------|
> | Offline computation cost             | 3284.05 ± 592.90 GPU-hours      |
> | Outputs storage offline              | 86.54 MB                        |
> | Signatures storage offline           | 400.00 KB                       |
> | Online computation cost              | 0.07 ± 0.01 GPU-hours           |
> | Outputs storage online               | 224.78 KB                       |
> | Signatures storage online            | 1.00 KB                         |
> | Direct evaluation computation cost   | 8.53 ± 1.54 GPU-hours           |
> | Computation savings wrt direct eval  | 8.46 ± 1.54 GPU-hours           |
> | Break-even point                     | 389 evaluations                 |
>
> As we can see, the majority of compute is spent on offline stage (3284 GPU-hours). Each DISCO-evaluation allows for saving (8.53 - 0.07 = 8.46 GPU-hours) per target model evaluation in comparison to direct evaluation. That means 389 (3284 / 8.46) target model evaluations are needed for DISCO to become more cost-effective than full evaluations.
>
> In practice, hundreds of target model checkpoint evaluations can naturally occur during model development. For example, a single OLMo 2 32B training run has 753 checkpoints on Hugging Face, already exceeding the break-even point.
>
> In some cases, one might not even need to spend compute on evaluating the source models and just download model predictions from open-source benchmarks like open-llm-leaderboard.

---

> ### Author Response · Authors · 2025-11-21
> **Response to W5**
>
> ## W5. Make the predictor adaptable to the number of anchor points
>
> We agree that adaptability to changing anchor set sizes is important in practice. The original submission already describes the use of an adaptive non-parametric predictor, kNN (see §4.2.2 in the submission). Non-parametric predictors, like kNN, adapt to different numbers of anchor points, as they do not require dedicated training for each feature dimension.
>
> To assess the adaptability of such predictors, we can analyse the kNN results on signatures that incrementally incorporate outputs on additional anchor points without retraining, as reported in Figure 5 of the submission. In those results, kNN performance slightly degrades from 0.97 to 0.95 in rank correlation when increasing the number of anchor points from 100 to 1000, suggesting it may not adapt well to a larger number of anchors.
>
> During the rebuttal, we hypothesized that the drop in kNN performance for a larger number of anchor points was caused by PCA. To check this, we recomputed the same results for kNN without PCA:
>
> | #anchors | Rank |
> |----------|------|
> | 1        | 54.3 |
> | 10       | 92.5 |
> | 100      | 96.2 |
> | 1000     | 96.9 |
> | 10000    | 97.5 |
>
>
> DISCO with kNN without PCA continues to perform well as more anchor points are added to the signatures, and performance even improves (e.g., Spearman correlation increases from 96.2 at 100 anchors to 96.9 at 1000).
> That demonstrates that DISCO with KNN naturally adapts to different signature sizes and varying output dimensionalities without retraining the predictor.

---

> ### Author Response · Authors · 2025-11-21
> **Response to Q1 + Q2 + Q3 + Q4**
>
> ## Q1 + Q2 + Q3 + Q4. Flaws in writing/table formatting
>
> Thank you, we fixed them in the revision (highlighted in blue text).

---

> ### Author Response · Authors · 2025-11-21
> **Response to Q6**
>
> ## Q6. More baselines for efficient evaluation in vision
>
> We agree that adding more baselines for efficient evaluation in the vision domain can strengthen the paper. We will respond to this question during the next week.

---

> > ### Comment · Reviewer_zKXy · 2025-11-26
> >
> > Thank you for the thorough and detailed responses. The authors have addressed the majority of my concerns, particularly regarding statistical rigor (confidence intervals and bootstrapped splits), computational cost analysis, and the adaptability of the predictor to varying anchor set sizes. I also appreciate the clarifications to the experimental setup and the fixes to the writing/formatting issues. However, I note that the failure analysis (W3 + Q5) remains incomplete, the performance-based split did not reveal failure modes, and the authors have indicated they will conduct further experiments with stronger distribution gaps. Additionally, the question of vision baselines (Q6) has not yet been addressed. Given that the authors have indicated they will be updating the manuscript to incorporate the new results and analyses from the rebuttal (for both new and pending experiments), I will gladly increase my score from a 6 to an 8 once the revised manuscript reflects these changes and the remaining analyses (failure modes under stronger distribution shift, vision baselines) have been provided. Please notify me when these updates are complete and I will verify accordingly.

---

> ### Author Response · Authors · 2025-11-27
> **W3 + Q5 and Q6 part. 2**
>
> Thank you for your response and for acknowledging our answers.
>
> ## W3 + Q5 part. 2
>
> We added experiments with a wider performance gap between source and target models to identify potential failure modes for DISCO. Inspired by [a], we introduce a performance split with varying gaps. We sort all models by their average performance and take top-30% (128 models) as target models, while using bottom-50% (213 models) as source models. The accuracy gap between the strongest source model and the weakest target model is 8.1%p.
>
> All model splits are summarized in the following table:
>
> |  | IID | Chronological | Performance w/o gap | Performance w/ gap |
> |---|---|---|---|---|
> | **Preliminary model sorting** | – | By timestamp | By performance | By performance |
> | **Source models** | All but target models | Bottom-90% | Bottom-90% | Bottom-50% |
> | **Target models** | Every 10th model | Top-10% | Top-10% | Top-30% |
>
> The table below reports Spearman’s rank correlation.
>
> |  | IID | Chronological | Performance w/o gap | Performance w/ gap |
> |---|---|---|---|---|
> | **Direct eval on random subset** | 92.1 ± 1.3 | 91.6 ± 2.6 | 89.8 ± 5.9 | 87.4 ± 5.7 |
> | **DISCO** | 98.6 ± 0.3 | 98.7 ± 0.2 | 98.1 ± 0.2 | 89.2 ± 1.0 |
> | **Mean difference (DISCO − direct eval)** | +6.5 | +7.1 | +8.7 | +1.8 |
>
> For source/target split with a performance gap, the difference between DISCO and direct evaluation is 1.8%p, which is significantly lower than for the IID split (6.5%p) or chronological split (7.1%p) or the performance split without a gap (8.7%p). We thus conclude that DISCO does break when the source and target model distributions differ, but only when the difference is unrealistically substantial (Performance w/ gap).
>
> We also discuss tasks that are not suitable for DISCO. The main constraint is that DISCO requires predictive probabilities for several predefined answer choices for each question. Answer choices correspond to the classes in Proposition 1 in the original submission. That makes DISCO not suitable for open-ended generation tasks such as translation or summarization. Applying DISCO to such tasks would first require defining sets of correct and incorrect outputs. We leave such experiments for future work.
>
> We have added this discussion to the limitations section and section F of the revised manuscript.
>
> [a] Zhang, Guanhua, Florian E. Dorner, and Moritz Hardt. *"How Benchmark Prediction from Fewer Data Misses the Mark."* NeurIPS Workshop (2025).
>
> ## Q6 part. 2
>
> We add the baseline results for the two closest methods: Lifelong Benchmark [a] (NeurIPS 2024) and SSEPY [b] (ECCV 2024).
>
> We computed the results for [a,b] ourselves, as the papers do not contain results on ImageNet. For fair comparison, we always select exactly **100 anchor points** from the ImageNet validation set and use the same set of models from the timm library [c]. We test on a chronological split of source and target models (350:50).
>
> | Method | Selection | Prediction | MAE ↓ | Rank ↑ |
> |--------|-----------|-------------|-------|---------|
> | **Baseline** | Random | Direct eval. | 3.03 | 65.2 |
> | **Lifelong Bench. [a]** | Source model correctness | Relative position to the hardest | 2.06 | 83.8 |
> | **SSEPY [b]** | Target model confidence | Horvitz-Thompson estimator | 3.05 | 76.2 |
> | **Model signature** | Random | KNN | 1.72 | 80.8 |
> | **Model signature** | Random | RF | 0.86 | 94.4 |
> | **DISCO** | High PDS | KNN | 1.68 | 81.9 |
> | **DISCO** | High PDS | RF | **0.63** | **96.9** |
>
> **DISCO (96.9 / 0.63) outperforms [a] (83.8 / 2.06) and [b] (76.2 / 3.05)** in rank correlation / MAE.
>
> We’ve updated the manuscript with these new results (blue text in Table 3 and Section I.2).
>
> [a] Prabhu, Ameya, et al. *"Lifelong benchmarks: Efficient model evaluation in an era of rapid progress."* NeurIPS 2024.
> [b] Fogliato, Riccardo, et al. *"A Framework for Efficient Model Evaluation through Stratification, Sampling, and Estimation."* ECCV 2024.
> [c] Wightman, Ross, et al. *"rwightman/pytorch-image-models: v0.8.10 dev0 Release."* Zenodo (2023).
>
> ## Incorporating new results from the previous answers in the manuscript
>
> We have also added results from W2 to section D and W4 to section B in the revised manuscript.

---

### Official Review · Reviewer_dg58 · 2025-10-30

**Soundness:** 3
**Presentation:** 4
**Contribution:** 3
**Rating:** 6
**Confidence:** 3

**Summary:**

This paper introduces DISCO (Diversifying Sample Condensation), an efficient evaluation approach that improves both steps common to prior frameworks: subset selection and performance prediction. Instead of clustering or relying solely on model confidence/correctness, DISCO selects samples that induce maximal inter-model disagreement, and then learns a direct mapping from a model signature—the concatenated raw outputs of a model on these selected samples—to predict full benchmark performance. Evaluated on 424 LLMs across MMLU, HellaSwag, Winogrande, and ARC, and 400 vision models on ImageNet-1k, DISCO achieves low MAE and high ranking fidelity with very small subsets (e.g., 100 samples), showing strong efficiency and competitive performance relative to prior methods.

**Strengths:**

- The paper identifies a clear and timely gap in the literature on efficient evaluation and proposes a simple, well-motivated solution that is both conceptually clear and practically effective (specially in vision domain and for low budget regimes).

- The mathematical framing provides valuable intuition for why the proposed approach works and strengthens the overall narrative without adding unnecessary complexity.

- The experimental evaluation is thorough and well executed, assessing the performance of the proposed method across multiple benchmarks and model families, and comparing it against several baselines.

- Table 2 (factor analysis) offers additional insight into how specific design choices affect performance.

- The paper is well written and structured, making the methodology and results easy to follow.

**Weaknesses:**

While the paper is clearly executed and well motivated, I have some reservations about the practical efficiency of the proposed approach, particularly for LLM evaluation.

- This first point is more of a critique of the overall line of work rather than one targeted specifically at DISCO. From Figure 5, it appears that randomly selecting slightly over 1000 samples with direct evaluation achieves on-par or even slightly better performance on both MAE and rank correlation compared to more sophisticated sample-selection techniques. This raises the question of whether the added complexity of sample-selection mechanisms is justified. To identify the most informative samples, one must evaluate a large pool of models, store their outputs, and compute disagreement statistics between their predictions for each sample. This process seems computationally and memory intensive, potentially offsetting the efficiency gains claimed from evaluating fewer samples. In contrast, simply randomly selecting a slightly larger subset (e.g., 2 000 samples) and then combining it with the proposed model-signature representation and simple predictors (e.g., linear or random forest) might yield even better results with significantly lower overhead.

- Relatedly, Table 1 shows that while the proposed method consistently improves over baselines across datasets, the magnitude of improvement in MAE appears modest—though I am open to being proven wrong, as this may depend on implementation details or evaluation scale. The improvement in rank fidelity is clear but numerically small given the additional computation required.

- Moreover, the paper does not include a quantitative analysis of computation or memory cost across methods. While this may not yet be standard practice in this research line, such an analysis would substantially strengthen the claim of “efficiency” and help contextualize the trade-off between computational expense and predictive accuracy.

- Finally, while the vision results (Table 3) are particularly strong and show a clear improvement in ranking fidelity, it remains unclear whether this work is the first to apply subset-based evaluation techniques to vision or if comparable baselines exist. Clarifying this point would help position the contribution more clearly within the broader literature.

I would like to note that I am open to revising my assessment if the authors can address these concerns. I appreciate the quality of the work and would genuinely welcome clarification that changes my current view.

**Questions:**

Please see the weaknesses section for a more detailed explanation of my concerns, but here are a few specific questions:

- Regarding the vision experiments: could you clarify whether this is the first application of subset-based evaluation techniques to vision models (e.g., ImageNet-1k), or if there are existing baselines in that domain that you compared against or were inspired by?

- About calibration: since the proposed method relies on model output probabilities both for computing disagreement (PDS) and constructing model signatures, how sensitive is it to differences in model calibration?

- On ranking fidelity: how should we interpret the magnitude of change in rank correlation you report? For instance, is improving from 0.916 (baseline) to 0.987 (your method on MMLU) a substantial and practically meaningful difference in ranking quality, or does it mainly indicate incremental improvement at the high end of the correlation scale?

---

> ### Author Response · Authors · 2025-11-21
> **Thank you + Response to W1**
>
> We appreciate the review and the comments on practical efficiency, calibration sensitivity, and the placement of the vision experiments.
> ## W1. Is the added complexity of sample-selection mechanisms justified?
>
> Thank you for the suggestion. We would be happy to initiate the change in the research practice in this domain, following your advice.
>
> We report computational costs of the main DISCO stages and compare them between the full DISCO pipeline and a simple baseline of “Model signature” (which we already introduced in Table 1 of the submission). The numbers are computed for a single H100 GPU and are extrapolated from the evaluation of 5 diverse 32B LLMs on MMLU (standard deviations across 5 runs).
>
> First, we remind the DISCO pipeline, which consists of two stages: an offline stage run once to set up DISCO, and an online stage run whenever a new target model is evaluated.
>
> - Offline
>   - Evaluate M source models on the full test dataset (M=385 in our experiments)
>   - Store source models' outputs from the previous step
>   - Select 100 anchor points that maximise PDS/JSD
>   - Concatenate outputs on anchor points to get model signatures
>   - Train a predictor that estimates model performance on the full test dataset from model signatures
> - Online
>   - Evaluate 1 target model on 100 anchor points
>   - Store the target model output from the previous step
>   - Concatenate outputs to get the target model signature
>   - Run the predictor to estimate the target model’s performance on the full test dataset
>
> For each target model, we reuse the same anchor points and the predictor obtained in the offline stage.
>
> | Item                                                | Value                      |
> |-----------------------------------------------------|----------------------------|
> | Full DISCO offline cost                             | 3284 ± 593 GPU-hours       |
> | “Model signature” on 2000 samples offline cost      | 468 ± 84 GPU-hours         |
> | “Model signature” gain in offline cost              | 2816 ± 593 GPU-hours       |
> | Full DISCO online cost                              | 0.07 ± 0.01 GPU-hours      |
> | “Model Signature” for 2000 samples online cost      | 1.21 ± 0.22 GPU-hours      |
> | Full DISCO gain in online cost                      | 1.14 ± 0.22 GPU-hours      |
> | Break-even point wrt “Model signature”              | 2471 evaluations           |
>
> Full DISCO-evaluation uses more compute than “Model signature” in the offline stage (2816 GPU-hours) while using less compute for each target model evaluation (0.07 GPU-hours vs 1.21 GPU-hours). That means 2471 (2816 / 1.14) target model evaluations are needed for DISCO to become more cost-effective than “Model signature”.
>
> In practice, hundreds of target model checkpoint evaluations can naturally occur during model development. For example, a single OLMo 2 32B training run has 753 checkpoints on Hugging Face. 3 training runs like this are enough to exceed the break-even point.
>
> In some cases, one might spend no compute on evaluating the source models by downloading the model predictions from open-source benchmarks like open-llm-leaderboard.
>
> We will include these results in the revised manuscript during the next week.

---

> ### Author Response · Authors · 2025-11-21
> **Response to W2 + Q3**
>
> ## W2 + Q3. Modest improvement in MAE and Spearman's rank correlation
> We agree that MAE improvements are less pronounced than rank improvements, but still believe they are significant. The only two places in Table 1 where the other method has better MAE than DISCO are: 0.86 for DISCO vs 0.80 for Metabench on Hellaswag, and 1.47 for DISCO vs 1.14 for Metabench on Arc. Metabench uses more anchor points for these datasets: 150 for Arc (50% increase) and 450 for Hellaswag (350% increase). Otherwise, it does not converge at all, as stated in the Table 1 caption in the submission. That means that for 100 anchor points, Metabench results are not applicable, and DISCO is the best overall in terms of both MAE and rank fidelity in that regime.
>
> The improvement in rank correlation that comes at a cost of additional computation is meaningful in terms of qualitative ranking improvement despite appearing numerically small (0.916 for direct evaluation vs 0.987 for DISCO on MMLU in Table 1 of the submission). We can see this by qualitatively comparing ground truth model rankings with rankings predicted by direct evaluation and DISCO. For this purpose, we include scatter plots of true vs. predicted ranks (https://drive.google.com/file/d/1qRS6Zk6Yt7pkRfDxbq9W5UHM0j8J3dQt/view?usp=sharing). The direct-evaluation predictor shows a noticeable spread around the diagonal, while DISCO’s predictions lie nearly perfectly on it.
>
> We will add this analysis to the revised manuscript during the next week.

---

> ### Author Response · Authors · 2025-11-21
> **Response to W3**
>
> ## W3. Quantitative analysis of computation cost across methods
>
> We agree that this type of analysis is highly valuable for practitioners. We report the amount of compute needed by each method during the main stages of the efficient evaluation pipeline. The experimental setup is identical to W1.
>
> | Approach             | Selection    | Prediction       | Offline compute (GPU-hours) | Online compute (GPU-seconds) |
> |----------------------|--------------|------------------|------------------------------|-------------------------------|
> | Baseline full eval   | -            | Direct evaluation| -                            | 30739 ± 5514                  |
> | Baseline             | Random       | Direct evaluation| -                            | 218 ± 39                      |
> | tinyBenchmarks       | Random       | gp-IRT           | 3284 ± 592                   | 219 ± 39                      |
> | tinyBenchmarks       | Anchor-corr  | gp-IRT           | 3284 ± 592                   | 219 ± 39                      |
> | tinyBenchmarks       | Anchor-IRT   | gp-IRT           | 3284 ± 592                   | 219 ± 39                      |
> | DISCO                | High PDS     | RF               | 3284 ± 592                   | 218 ± 39                      |
> | DISCO                | High PDS     | KNN              | 3284 ± 592                   | 218 ± 39                      |
>
>
> The differences in method online costs are, e.g., 219 - 218 = 1 GPU-second for offline tinyBenchmarks vs. DISCO, which is negligible compared to the time required for target model evaluation on anchor points (218). For offline costs, the difference becomes zero after rounding.
> We will include these results in the revised manuscript during the next week.

---

> ### Author Response · Authors · 2025-11-21
> **Response to W4 + Q1**
>
> ## W4 + Q1. Subset-based evaluation techniques in vision
>
> Thank you for the suggestion. We will respond to this question during the next week.

---

> ### Author Response · Authors · 2025-11-21
> **Response to Q2**
>
> ## Q2. How sensitive is DISCO to differences in model calibration?
>
> To evaluate the sensitivity of DISCO to model calibration, we compared the Expected Calibration Error (ECE) of target models with the Mean Absolute Error (MAE) between their true performance and DISCO-predicted performance on MMLU. We observe a Pearson correlation of 0.49 between ECE and MAE, indicating that better-calibrated models lead to more accurate performance predictions by DISCO. To understand why that happens, let's consider two extreme cases of calibration. For a perfectly calibrated model, the relationship between prediction confidence and correctness is deterministic and monotonic, which implies high mutual information between the two. In contrast, for a highly miscalibrated model (e.g., random guessing or uniformly confident but wrong), prediction confidence becomes almost statistically independent of correctness, resulting in low mutual information. Consequently, the more calibrated a model is, the more predictive its confidence patterns are of its true performance, and therefore the more informative its signature is for performance prediction.
>
> The corresponding MAE vs ECE scatter plot is available at the following link: https://drive.google.com/file/d/1sECNHIOovVF3VRiKKWe1i8wSfuSZA7gY/view?usp=sharing
>
> In the process, we realized that two factors are confounded in calibration: overall model confidence and how well predictive uncertainty is reflected in the confidence estimate. To isolate overall confidence, we compare MAE with mean confidence separately. We observe a Pearson correlation of -0.47 between MAE and confidence, indicating that the overall confidence level is the main factor of ECE that matters for DISCO performance.
>
> The scatter plot for MAE vs confidence: https://drive.google.com/file/d/1IZnaU6a63jLBSwoSD_8leH7RAFRjUyKy/view?usp=sharing
>
> We will add these results to the revised manuscript next week.

---

> ### Author Response · Authors · 2025-11-27
> **W4 + Q1. Subset-based evaluation techniques in vision (part. 2) and adding results previous responses to the manuscript.**
>
> Thank you for initiating this discussion. Our work is **not the first** to propose efficient evaluation in the vision domain. The two closest methods are **Lifelong Benchmark [a]** (NeurIPS 2024) and **SSEPY [b]** (ECCV 2024). They propose efficient evaluation methods for visual models, using a similar **two-stage framework** to efficient evaluation methods in the language domain (see §4 / Fig. 2 of our submission):
>
> **Stage 1:** Select *“important/representative”* anchor points
> **Stage 2:** Estimate model performance based on the model outputs on the anchor points
>
> Our main message is that **for selecting anchor points for efficient evaluation**, it is better to select **diversity-inducing data points (our DISCO)** than to **make a good coverage of sample difficulty** (prior work). In our submission, we show that our approach **outperforms** the prior approaches.
>
> Likewise, in the vision domain, the existing approaches [a, b] focus on good coverage of sample difficulty, rather than on maximizing per-sample information by seeking diversity-inducing data points. We test empirically whether the same message holds in the vision domain by comparing DISCO to [a] and [b]. As in Table 3 in the submission, **MAE** is the mean absolute error, and **Rank** indicates Spearman’s rank correlation.
>
> We computed the results for [a, b] ourselves, as the papers do not contain results on ImageNet. For fair comparison, we always select exactly **100 anchor points** from the ImageNet validation set and use the same set of models from the timm library [c]. We test on a chronological split of source and target models (350:50).
>
> ---
>
> | Method | Selection | Prediction | MAE ↓ | Rank ↑ |
> |--------|-----------|-------------|-------|---------|
> | **Baseline** | Random | Direct eval. | 3.03 | 65.2 |
> | **Lifelong Bench. [a]** | Uniform correctness | Relative position to the hardest | 2.06 | 83.8 |
> | **SSEPY [b]** | Uniform confidence | Horvitz-Thompson estimator | 3.05 | 76.2 |
> | **Model signature** | Random | KNN | 1.72 | 80.8 |
> | **Model signature** | Random | RF | 0.86 | 94.4 |
> | **DISCO** | High PDS | KNN | 1.68 | 81.9 |
> | **DISCO** | High PDS | RF | **0.63** | **96.9** |
>
> ---
>
> **DISCO (96.9 / 0.63) outperforms [a] (83.8 / 2.06) and [b] (76.2 / 3.05)** in rank correlation / MAE.
> The conclusion from language experiments still holds: instead of selecting anchor points with uniform coverage of sample difficulty measured by model correctness scores or confidences, **one should focus on selecting the points on which models typically disagree.**
>
> We’ve updated the manuscript with these new results (blue text in Table 3 and Section I.2).
>
> ---
>
> [a] Prabhu, Ameya, et al. *"Lifelong benchmarks: Efficient model evaluation in an era of rapid progress."* NeurIPS 2024.
> [b] Fogliato, Riccardo, et al. *"A Framework for Efficient Model Evaluation through Stratification, Sampling, and Estimation."* ECCV 2024.
> [c] Wightman, Ross, et al. *"rwightman/pytorch-image-models: v0.8.10 dev0 Release."* Zenodo (2023).
>
> ## Incorporating new results from the previous answers in the manuscript
>
> We have also added results from W1 to section B, from W2+Q3 to section C, from W3 to section B, and from Q2 to section E in the revised manuscript.

---

### Official Review · Reviewer_W8ah · 2025-10-31

**Soundness:** 3
**Presentation:** 3
**Contribution:** 4
**Rating:** 8
**Confidence:** 3

**Summary:**

This paper sheds light on a important problem of efficiently estimating a model's performance of large contemporary benchmarks $D$ by evaluating the model performance on a much smaller selected subset $|A| = K \ll D$. The core idea, DISCO, puts forward the notion that samples should be selected where models disagree the most (to maximize inter-model response diversity), not necessarily samples that are "representative" in input latent space. This is done using either Jensen-Shannon divergence (JSD) or a computationally simpler proxy, Predictive Diversity Score (PDS).

After selecting $A$ each model is represented by its "signature" which is then mapped using a lightweight predictor (k-NN/Random Forest/Regression) to estimate actual dataset performance. The paper also provides theoretical justification for using PDS by sandwiching bounds with JSD and connecting it with mutual information.

The empirical performance is strong across language datasets like MMLU, HellaSwag, ARC, Winogrande and also in Vision (Imagenet) while dramatically reducing inference cost with $K = 100$.

**Strengths:**

1. The insight of shifting focus from representativeness of samples to disagreement of models on these samples is a novel and fresh perspective.
2. PDS is computationally cheap to calculate from logits and the final predictor from the model signature is straightforward to implement in existing benchmark evaluation pipelines.
3. The authors provide reasonable theoretical bounds to use PDS as a proxy for JSD for the subset dataset selection.
4. The contribution includes strong empirical results and generality across domains, which adds to the fact that this method is model agnostic, thus is more likely to be used in the real world.

**Weaknesses:**

1. In proposition 1, the authors implicitly use the assumption - uniform prior over source models. This is not examined fully in practice because real model pools are heterogeneous and non-uniform. How sensitive are the PDS/JSD selection and signature predictors to such non-uniformities?
2. Computing PDS/JSD requires running $M$ source models across the entire benchmark dataset $D$ which is likely significant for large datasets. The authors should report absolute compute (GPU-hours), memory/storage for signatures and break-even points (how many target models or evals justify the offline cost) for the method to be more practically used.
3. Minor typo in section 4.2.2. "we also compare against a rather complex prior work
and show that our simple method wins against the stolen" - The word stolen is out of context.

**Questions:**

See weaknesses

---

> ### Author Response · Authors · 2025-11-21
> **Thank you + Response to W1.1 and W1.2**
>
> Thank you for the thoughtful and constructive review. We appreciate your detailed comments on the theoretical framing, empirical results, and practical relevance of our approach.
>
> ## W1.1 DISCO assumes uniform prior over source models
> Proposition 1 does not assume a uniform prior over source models $\{f^1, \ldots, f^M\}$. It only assumes that the model index is drawn uniformly $m \sim \text{Unif}${$1, \ldots, M$}. If the prior over models is non-uniform, we can replicate models proportionally to their weights so that the index distribution becomes uniform without changing the underlying model behaviour.
>
> ## W1.2 PDS/JSD selection and predictor sensitivity to source models heterogeneity
> PDS/JSD selection directly depends on source models heterogeneity. Model heterogeneity can be characterized by how much each model’s output entropy $H(f^m(x))$ deviates from the entropy of the mean distribution $H\left(\frac{1}{M}\sum_m f^m(x)\right)$. The larger this deviation, the higher the JSD. According to Proposition 1, larger JSD is equivalent to higher mutual information between outputs and benchmark accuracy. That leads to better performance of the DISCO predictor. Conversely, if there is no deviation in entropies, for example, many copies of the same model, then JSD as well as mutual information become zero. That leads to poor performance of the predictor.

---

> ### Author Response · Authors · 2025-11-21
> **Response to W2.1, W2.2 and W3**
>
> ## W2.1 DISCO computational cost report
>
> Thank you for the suggestion. We report space-time complexity for the main stages of DISCO as well as the cost of direct evaluation of the target model. The numbers are computed for a single H100 GPU and are extrapolated from the evaluation of 5 diverse 32B LLMs on MMLU (standard deviations across 5 runs).
>
> First, we remind the DISCO pipeline, which consists of two stages: an offline stage run once to set up DISCO, and an online stage run whenever a new target model is evaluated.
>
> - Offline
>   - Evaluate M source models on the full test dataset (M=385 in our experiments)
>   - Store source models' outputs from the previous step
>   - Select 100 anchor points that maximise PDS/JSD
>   - Concatenate outputs on anchor points to get model signatures
>   - Train a predictor that estimates model performance on the full test dataset from model signatures
> - Online
>   - Evaluate 1 target model on 100 anchor points
>   - Store target model output from the previous step
>   - Concatenate outputs to get the target model signature
>   - Run the predictor to estimate the target model’s performance on the full test dataset
>
> For each target model, we reuse the same anchor points and the predictor obtained in the offline stage.
>
>
> | Item                                   | Value                             |
> |----------------------------------------|-----------------------------------|
> | Offline computation cost               | 3284.05 ± 592.90 GPU-hours        |
> | Outputs storage offline                | 86.54 MB                          |
> | Signatures storage offline             | 400.00 KB                         |
> | Online computation cost                | 0.07 ± 0.01 GPU-hours             |
> | Outputs storage online                 | 224.78 KB                         |
> | Signatures storage online              | 1.00 KB                           |
> | Direct evaluation computation cost     | 8.53 ± 1.54 GPU-hours             |
> | Computation savings wrt direct eval    | 8.46 ± 1.54 GPU-hours             |
>
>
> We can see that the majority of compute is spent on offline stage (3284 GPU-hours).
>
> ## W2.2 How many DISCO-evaluations justify its one-time set-up cost?
> DISCO breaks even against the direct evaluation baseline at 389 evaluations. In the table from W2.1, each DISCO-evaluation allows for saving 8.46 GPU-hours (8.53 - 0.07) per target model evaluation in comparison to direct evaluation (8.53 GPU-hours). It sets the break-even point at 389 (3284 / 8.46).
>
> In practice, hundreds of target model checkpoint evaluations can naturally occur during model development. For example, a single OLMo 2 32B training run has 753 checkpoints on Hugging Face, already exceeding the break-even point.
>
> In some cases, one might spend no compute on evaluating the source models by downloading the model predictions from open-source benchmarks like open-llm-leaderboard.
>
> To illustrate the computational benefits of efficient evaluation methods, we report the amount of computation required by alternative approaches to estimate performance.
>
> | Approach              | Selection    | Prediction       | Offline compute (GPU-hours) | Online compute (GPU-seconds) |
> |-----------------------|--------------|------------------|------------------------------|-------------------------------|
> | Baseline full eval    | -            | Direct evaluation| -                            | 30739 ± 5514                  |
> | Baseline              | Random       | Direct evaluation| -                            | 218 ± 39                      |
> | tinyBenchmarks        | Random       | gp-IRT           | 3284 ± 592                   | 219 ± 39                      |
> | tinyBenchmarks        | Anchor-corr  | gp-IRT           | 3284 ± 592                   | 219 ± 39                      |
> | tinyBenchmarks        | Anchor-IRT   | gp-IRT           | 3284 ± 592                   | 219 ± 39                      |
> | DISCO                 | High PDS     | RF               | 3284 ± 592                   | 218 ± 39                      |
> | DISCO                 | High PDS     | KNN              | 3284 ± 592                   | 218 ± 39                      |
>
>
> The differences in method online costs are, e.g., 219 - 218 = 1 GPU-second for offline tinyBenchmarks vs. DISCO, which is negligible compared to the time required for target model evaluation on anchor points (218). For offline costs, the difference becomes zero after rounding.
>
> During the next week, we will add these results to the revised manuscript.
>
> ## W3. Minor typo
> Thanks. We fixed the typo in the revision (highlighted in blue text).

---

> > ### Comment · Reviewer_W8ah · 2025-11-26
> >
> > The authors have addressed the major concerns raised, including some overlapping questions from other reviewers. My rating thus remains unchanged, and I recommend accepting the paper.

---

> ### Author Response · Authors · 2025-11-27
> **Thank you**
>
> Thank you for your response, for acknowledging our answers, and for a high score.
>
> We have also added the results from W2 to section B in the revised manuscript.

---

### Author Response · Authors · 2025-12-02
**Rebuttal summary for AC part 1**

Dear AC, we provide below summaries of paper contributions, initial reviews, author-reviewer discussions, and implemented improvements.

## Summary of paper contributions

We propose a new efficient evaluation method, Diversifying Sample Condensation (DISCO). It estimates full evaluation results on MMLU at only 0.07% cost (see §B.4 in the revised manuscript), reconstructing ground truth model ranking with up to 0.987 Spearman rank correlation (Table 1). Our method uses the same two-stage framework as prior methods:

**Stage 1:** Select "important/representative" anchor data points from the full test set based on the source models’ outputs on them.

**Stage 2:** Estimate the target model's performance based on its outputs on the anchor points.

Stage 1 needs to be run only once, while Stage 2 is run each time a new target model needs to be evaluated.

For the first stage, prior work selected anchor points to cover the full range of sample difficulty. For the second stage, they applied weighted sums of correctness scores on those anchor points to predict final model performance.

DISCO, in contrast, selects diversity-inducing data points in the first stage. In the second stage, DISCO models the mapping between source-model outputs and performance.

We verify our improvements both theoretically and empirically.

## Summary of initial reviews

**Reviewer 1 (W8ah)** gave a **score of 8**. They asked for clarification regarding the sensitivity of DISCO to the heterogeneity of models (R1.W1). Our method involves both offline cost (Stage 1) and online cost (Stage 2). They asked whether the offline cost is justified by the online cost savings and requested clarification of the break-even point (R1.W2).

**Reviewer 2 (dg58)** gave a **score of 6** and indicated that they are ready to **increase the score if all the mentioned weaknesses are addressed**. They requested details on DISCO’s compute and memory costs in comparison to other methods (R2.W1 & R2.W3). They asked to clarify DISCO’s relative position with respect to other subset-based evaluation methods in the vision domain (R2.W4+Q1). They suggested an analysis of the influence of model calibration on DISCO’s performance (R2.Q2).

**Reviewer 3 (zKXy)** gave a **score of 6** and indicated that they are willing to **raise the score once the identified weaknesses are fully resolved**. They asked to provide additional results for statistical rigor (R3.W2), more challenging train/test splits (R3.W3+Q5), and more vision baselines (R3.Q6). They also asked to share details on computational costs (R3.W4).

---

> ### Author Response · Authors · 2025-12-02
> **Rebuttal summary for AC part 2**
>
> ## Summary of discussions
>
> **Reviewer 1 (W8ah)** acknowledged that we addressed their concerns and left the rating of 8 unchanged.
>
> **Reviewer 2 (zKXy)** has not responded to our rebuttal. Since we addressed every concern they raised, we believe they would increase the score if they had the opportunity. We think so because they explicitly indicated their openness to revising their assessment.
>
> **Reviewer 3 (zKXy)** acknowledged that our intermediate response addressed most of their concerns. They said they would increase the score from 6 to 8 once we published the remainder of our response with more challenging train/test splits and new baselines for the vision domain. We published the respective response later that day and believe it clarified all remaining concerns. The reviewer response form became uneditable right after this exchange.
>
> ## Summary of improvements
>
> Thanks to the discussion, the paper became stronger by incorporating the following updates in the revised manuscript (highlighted in blue in the updated PDF):
>
> - Computational and memory costs compared against existing efficient evaluation methods (§B).
> - Qualitative results for improvements over baselines (§C).
> - Statistical rigor of the results (§D).
> - DISCO’s sensitivity to model calibration (§E).
> - Empirical analysis of DISCO’s performance on challenging train/test splits (§6 and §F).
> - New baselines for efficient model evaluation in the vision domain. Their results confirmed that the anchor points selection and performance prediction method introduced in DISCO excels in both vision and language domains (§5.4 and §I.2).
>
> Our response to each weakness.
>
> **Reviewer 1 (W8ah):**
>
> R1.W1: clarified that DISCO benefits from heterogeneous source models.
>
> R1.W2: reported space-time complexity for the main stages of DISCO as well as the cost of direct evaluation of the target model.
>
> R1.W3: fixed the typo in the revised manuscript.
>
> **Reviewer 2 (dg58):**
>
> R2.W1 & R2.W3: reported computational costs of the main DISCO stages and compared the compute usage of different efficient evaluation methods.
>
> R2.W2+Q3: clarified the magnitude of improvements in metrics over other methods; provided qualitative results.
>
> R2.W4+Q1: compared DISCO to new efficient evaluation baselines in the vision domain.
>
> R2.Q2: examined how model calibration relates to the error in DISCO’s performance predictions.
>
> **Reviewer 3 (zKXy):**
>
> R3.W1: clarified experimental setup in the revised manuscript.
>
> R3.W2.1 & R3.W2.2: shared confidence intervals for results, and computed additional results for different train/test splits.
>
> R3.W3+Q5: added empirical analysis to scenarios where DISCO fails and clarified DISCO’s limitations.
>
> R3.W4: reported computational costs of the main DISCO stages and compared the compute usage of different efficient evaluation methods.
>
> R3.W5: analysed the version of DISCO’s predictor from Stage 2 that is adaptable to a variable number of anchor points.
>
> R3.Q1+Q2+Q3+Q4: fixed flaws in writing/table formatting in the revised manuscript.
>
> R3.Q6: compared DISCO to new efficient evaluation baselines in the vision domain.
>
> ## Concluding remark
>
> The review process has solidified our contribution with additional results and discussions. The paper originally received scores of 8, 6, and 6, with the last two reviewers expressing willingness to increase their scores once their concerns were addressed. We have addressed all the concerns raised by the reviewers. Reviewer 1 confirmed it. Reviewers 2 and 3 would likely have raised their scores to 8 or higher if given the chance.

---

### Meta-Review · Area_Chair_eFAJ · 2026-01-07

**Summary:**

This is a clear accept as the reviews are universally positive. The reviewers like the high-level idea of selecting a subset of points for evaluation for models not based on the diversity/representativeness of the points themselves but rather on the diversity of model outputs on those points. They felt that the paper was clear and the evaluation was very comprehensive.

**Reviewer Concerns:**

One concern was that simple random sampling of say 1000-2000 datapoints can perform on par with DISCO, making it unclear that the added complexity of careful sample selection is worthwhile. Related to this, many reviewers pointed out that there is not enough discussion of the computational cost of DISCO. The authors do a good job of responding to this, showing running time costs for DISCO and identifying the break even point vs simple random subset selection, etc. I strongl encourage them to include this evaluation in the revised version. I also perhaps think some reviewers were looking for a more theoretical discussion of scalability -- i.e., how does the runtime scale with the number of source models, the size of the benchmark dataset, etc.

One reviewer mentioned that not enough baselines were included for the assessment on vision models -- in the rebuttal the authors add additional baselines.

**Reviewer Scores:**

Reviewers dg58 and zKXy perhaps would have raised their scores from 6 to 7 given the comprehensive responses to their concerns.

---

### Decision · Program_Chairs · 2026-01-26

Accept (Poster)